# Chromian Spinels from Kazanian-Stage Placers in the Southern Pre-Urals, Bashkiria, Russia: Morphological and Chemical Features and Evidence for Provenance

**Ildar R. Rakhimov [1,*], Dmitri E. Saveliev [1], Mikhail A. Rassomakhin [2] and Aidar A. Samigullin [1]**

1 Ufa Federal Research Center, Institute of Geology, Russian Academy of Sciences, 16/2Karl Marx Street, 450077 Ufa, Russia; sav71@mail.ru (D.E.S.); samigullinaidar85@gmail.com (A.A.S.)

2 South Urals Federal Research Center of Mineralogy and Geoecology of UB RAS, Ilmeny Reserve, 456300 Miass, Russia; miha_rassomahin@mail.ru

\* Correspondence: rigel92@mail.ru; Tel.: +7-919-159-0904

**Abstract:** Six minor alluvial chromite placers (Kolkhoznyi Prud, Verkhne-Yaushevo, Sukhoy Izyak, Bazilevo, Novomikhaylovka, Kiryushkino) and one major littoral placer (Sabantuy) were found in sandy sediments of the Kazanian stage of the Permian System (Late Roadian and Wordian Stages) in the Southern Pre-Urals. It is shown that the morphological features of chromian spinels are diverse, which is not evidence of the heterogeneity of the source. The bulk chemical composition of chromian spinels from all placers is similar and generally correlates with compositions of chromian spinels from the Kraka ophiolitic complex in the Southern Urals. The morphological diversity of grains, varied chemical composition and presence of melt inclusions in Ti-high octahedral grains of chromian spinels comply with the ophiolitic nature of the source. Thus, there is no need to refer to other sources for chromite ores but ophiolitic. The new placers expand the dissemination area of chromite-bearing deposits on the east edge of the East-European Platform and offer a prospect to discover new placers.

**Keywords:** Southern Pre-Urals; littoral and alluvial placers; detrital chromian spinel; morphogroup; ophiolitic source

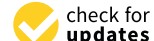



## 1. Introduction

Chromian spinel is a widespread mineral that occurs in different types of magmatic, metamorphic and sedimentary rocks [1–5]. However, the bulk of chromian spinels is associated with magmatic rocks [4,6–9]. In terrigenous sedimentary rocks, chromian spinel occurs as an important heavy mineral, which allows tracing the sediment source [10,11]. The indicative role of chromian spinel is associated with its varying chemical composition dependent on the magma composition and P-T-fO$_2$ conditions of crystallization [9,12–19]. High resistance of chromian spinel to supergene corrosion and mechanical wear determines its accumulation in placers [20,21].

The term "chromite" is commonly used to designate any Cr-rich mineral of the spinel group as well, but pure chromite is rare in nature. A more collective term is the chromian spinel, but for convenience chromian spinel placers are usually called chromite placers. Chromite placers are divided into three genetic types, i.e., colluvial, alluvial and littoral [22]. The former occurs close to the provenance (area of the Great Dike in Zimbabwe, Camaguey in Cuba, Sarany in the Ural region), while the rest are distal to the source (New Caledonia, the Hokkaido Island in Japan, the Maharashtra bank in India, the Oregon coast in the USA, the north-eastern coast of the RSA, etc.) [2,23–27]. The contribution of placers to the global chrome production is still minor (1.3%), but some comprise impressive chrome reserves (e.g., 60 mt in the Great Dike area in Zimbabwe with the placer area of 518 km$^2$ and a thickness of the ore bed of 0.5 m). In addition, there are distal littoral placers, where chromite is an accessory mineral associated with complex deposits, e.g., chrome-titanium-zirconium (Ukrainian Shield, Nizhny Novgorod region in Russia) [28]. In the Ural region,

chromite placers are known on the western (Sarany deposit) [29] and eastern (Alapay and Varshavka occurrences) slopes [30]. These placers are colluvial and occur as boulder accumulations. No littoral paleoplacers are known in the Ural region, while alluvial placers are minor and have no commercial value, even potentially.

Many works are devoted to the study of detrital chromian spinels. However, the study of their morphology is assigned a secondary role, or is not considered at all. Morphological diversity of chromian spinels is usually associated with the heterogeneity of the source. In this study, we show that morphological diversity is not necessarily related to source heterogeneity.

### 1.1. Crystallography of Chromian Spinel

Chromite ($FeCr_2O_4$) crystallizes in the isometric system. It is one of the spinel group minerals with the general formula $AB_2O_4$ that contains 24 cations (8 cations in the tetrahedral site and 16 cations in the octahedral site) and 32 oxygen atoms, where the oxygen atoms alternate with cations in a cubic closest packed arrangement [31,32]. Spinels can be structurally normal (A cations in tetrahedral sites, B cations in octahedral sites), inverse (B cations are evenly divided among the tetrahedral and octahedral sites), or partially inverse (B cations are unevenly divided among the tetrahedral and octahedral sites) [9,33]. The spinels are cubic, with the space group Fd3m. The chemical composition of chromian spinel, i.e., Cr-bearing and Cr-rich spinels (($Fe^{2+}$, Mg, $Ti^{4+}$, Mn) [Cr, Al, $Fe^{3+}$]$_2O_4$), varies. It correlates with the impact of chemical substitution on both the tetrahedral and octahedral sites with a unit cell parameter [33]. The unit cell parameter of spinel group minerals ranges from 8.080 to 8.536 Å [32]. In spinel group minerals, the unit cell parameter is mainly governed by composition and not by temperature and pressure of crystallization [34].

Chromite (chromian spinel) crystals typically occur as an octahedron, a rhombododecahedron and varied combinatorial polyhedra, but in parental rocks, chromite is commonly represented by xenomorphic grains [35–38]. The morphology of detrital chromian spinels is used as an indicator of their origin and evolution in the supergene zone [20,39–41]. The morphology of chromian spinels has been considered an important criterion for distinguishing diamond-bearing associations in the source for terrigenous deposits. It involves the presence of octahedral crystals with vicinals and myriohedral grains [20,40] and flattened octahedral crystals [42] in the heavy mineral fraction of alluvial sediments. Studies indicate that accessory chromian spinels in volcanic rocks typically show octahedral crystal forms [43–45]. In chromite ores of ophiolitic ultrabasites and concentrically zoned massifs, chromian spinel produces granular masses, ball-shaped segregations, i.e., nodules, where individual crystals can be hardly observed [36,37,46]. After erosion of these formations, chromian spinels become rounded clastic fragments or various xenomorphic grains [47]. In komatiites, ophiolites, ophiolitic chromitites and layered massifs, chromian spinel occurs both as idiomorphic octahedral and dodecahedral crystals and as amoeba-like and dendritic crystals [35,38,45,48–58].

### 1.2. Chemistry of Chromian Spinel as an Indicator of Magmatic Formations

Signatures of the chromian spinel chemical composition have been long used to reconstruct petrogenetic settings of the parental rock [12,13,15,59–63]. Chromian spinels of ophiolitic complexes show widely varied $Cr_2O_3$ and $Al_2O_3$ contents, due to an almost total $Cr^{3+} \leftrightarrow Al^{3+}$ isomorphic substitution, low $Fe^{3+}$ and $TiO_2$ contents and a broadly ranged Mg content (Mg# = 0.4–0.8) [60,61]. Ophiolites include chromite, alumochromite, chrompicotite and picotite mainly, while chromitites from ophiolites commonly contain alumochromite and chromite, i.e., spinels richest in Cr [38,64,65]. Mid-ocean ridge basalts (MORB) and peridotites show a narrower range of $Cr_2O_3$ and $Al_2O_3$, while the $Al_2O_3$ and $Fe^{3+}$ content is higher compared to ophiolites [59,60]. Besides, the latter are inherently marked by high Mg content of chromian spinels (Mg# = 0.6–0.8) [59]. Chromian spinels in concentrically zoned platinum-bearing and layered massifs typically show a wide spectrum of $Cr^{3+} \leftrightarrow Fe^{3+}$ isomorphic substitutions and thus widely distributed ferrichromite and chrommagnetite [60].

Chromian spinels from diamond-bearing kimberlites are the highest in $Cr_2O_3$ (>62 wt.%), high in Mg (Mg# = 0.7–0.8), but low in $Al_2O_3$ (<7.5 wt.%) and $TiO_2$ (<0.5 wt.%) [40]. In island-arc volcanites, in particular, those of primitive island arcs, and in ocean island basalts, accessory chromian spinels are high in Cr (Cr# = 0.7–0.9) and vary in the Mg content (Mg# = 0.2–0.8) [59,60]. In general, chromian spinels from MOR volcanites, island arcs and oceanic islands are high in $TiO_2$ (0.5–2 wt.% and more) [59].

All these signatures of the chromian spinel geochemistry made it possible to elaborate discrimination diagrams to distinguish magmatic formations. The most popular among these diagrams are $Al^{3+}$–$Cr^{3+}$–$Fe^{3+}$, Cr#–Mg#, Cr#–Fe#, $Al_2O_3$–$TiO_2$, $TiO_2$–$Fe^{2+}/Fe^{3+}$ [60,61]. Analysis of geochemical signatures of detrital chromian spinels using these discrimination diagrams is a critical tool for interpretation of sources for terrigenous sediments [21,66–68].

## 2. Geological Background

The Hercynian orogeny stage is considered one of the key stages in the history of the Uralian Fold Belt [69,70]. During the Permian period, the typical tectonic structure of the Urals with its submeridional orientation was formed. The geological structure of the Southern Urals is represented by blocks of the continental and oceanic crust of the Precambrian, Early and Late Paleozoic age. The most interesting among these are island-arc complexes of the Eastern slope and ophiolitic complexes of the Western slope [71]. The Kraka massif is one of largest and best-studied ophiolitic complexes in the Urals. It is divided into four conjugated massifs, i.e., the Southern, Middle, Northern and Uzyansky Kraka [72]. The origin of the Kraka massif has been debated. Currently, the most popular concept suggests its allochthonous bedding, since the massif is considered a fragment of an ophiolitic nappe [71]. In general, magmatic rocks are poorly distributed on the Western slope of the Southern Urals, while Precambrian and Paleozoic sedimentary and sedimentary-metamorphogenic complexes prevail [69].

The Southern Pre-Ural region is currently considered a margin of the East-European Platform (EEP), where separate blocks with the uplifted or sunk basement are defined, i.e., the Buzuluk Depression, Southern Dome of the Tatar Arch and Birskaya Depression [71]. The eastern part of the Southern Pre-Urals is marked by the Ural Foredeep that stretches along the whole Urals structure and contains Early Permian mainly terrigenous and evaporitic sediments [69,73]. The upper sedimentary cover of EEP is composed of Middle Permian sediments in almost all of the Southern Pre-Urals. Among the sediments, those of the Kazanian stage (corresponding with the Late Roadian and Wordian in International Chronostratigraphic Chart) [21] deserve particular attention. The Sabantuy chromite paleoplacer has been recently found in Kazanian-stage sandy sediments in the Southern Pre-Urals. Ophiolitic and volcanogenic complexes of the Southern Urals are considered the main source for this placer [21,74].

The discovery of the Sabantuy paleoplacer attracted close attention to the Kazanian sandy sediments in the Pre-Urals [28]. Our research discovered several minor paleoplacers were that cluster in sandy-pebble sediments of the Kazanian stage, i.e., the Kolkhoznyi Prud, Verkhne-Yaushevo, Sukhoy Izyak, Bazilevo, Novomikhaylovka and Kiryushkino placers. The new placers are located at the distance of 5 to 25 km from the Sabantuy paleoplacer (Figure 1). All of them were found in quarries, where sandy-gravel material was mined. The current paper provides research data on morphological and chemical features of chromian spinel grains from each of the seven sections mentioned. Discussed below are geological and lithological signatures of chromite-bearing sections, conditions of their accumulation and the indicative role of the chromian spinel morphology and composition in reconstruction of their provenance.

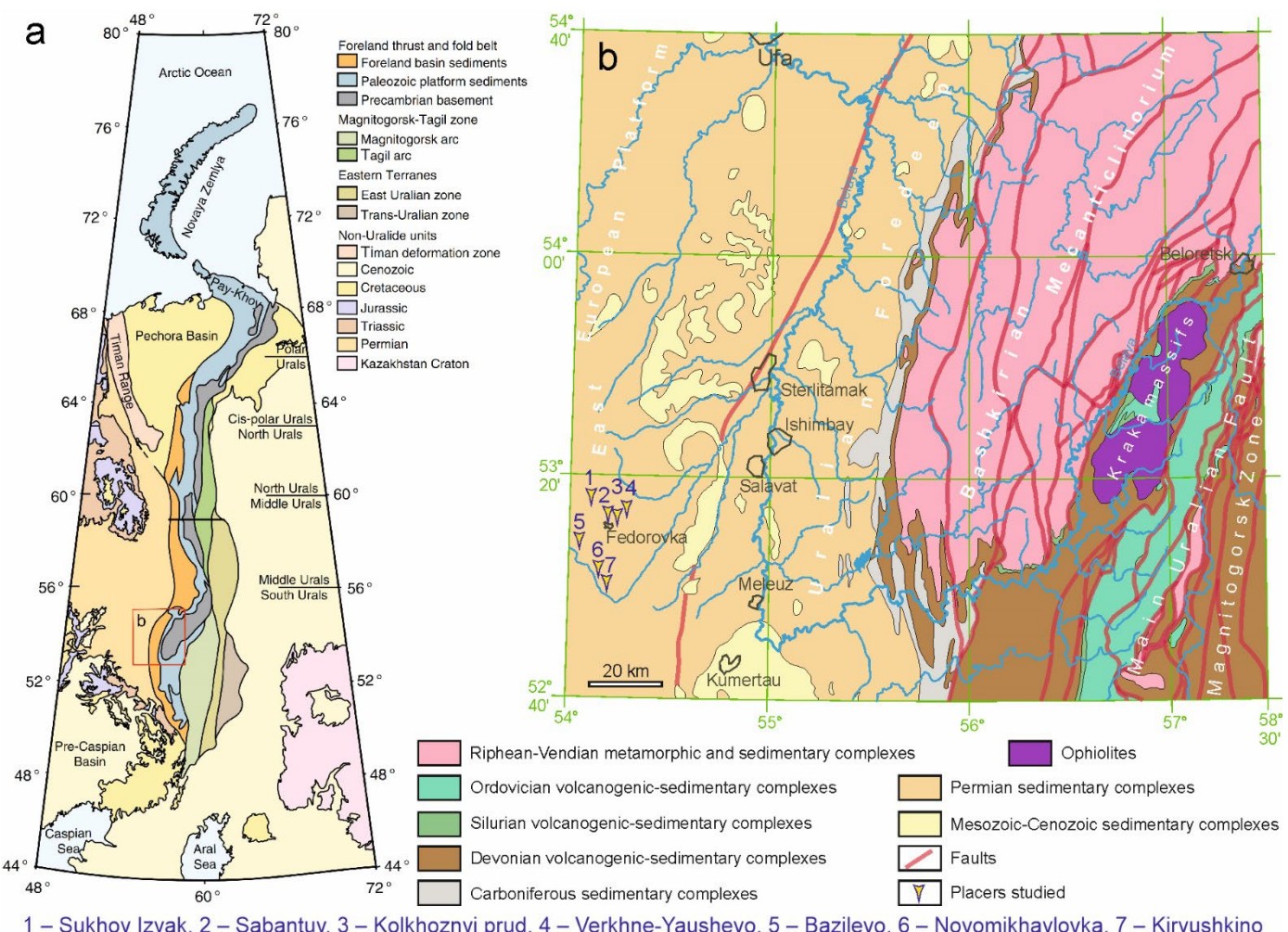

**Figure 1.** Structural zones of the Uralides [70] (**a**) Corresponding fragment of the State geological map N-40 (Ufa) simplified after Knyazev et al. [75] with the location of the studied chromite placers (**b**).

The main aim of this article is to find the relationship between the morphological features of detrital chromian spinels and their chemical composition.

## 3. Materials and Methods

Twelve sandstone samples from seven outcropped sections of chromite-bearing sandy sediments were studied in this research, i.e., four samples from the Sabantuy paleoplacer, three samples from the Kiryushkino paleoplacer and one sample each from the Verkhne-Yaushevo, Kolkhoznyi Prud, Novomikhaylovka, Bazilevo and Sukhoy Izyak paleoplacers. Chromian spinel grains were originally picked out using a binocular microscope for further survey under the electron microscope. A detailed petrographic study was provided for 25 samples of relatively firm-cemented sandstones from chromite-bearing sediments on Carl Zeiss Axioskop 40 A optic microscope (Zeiss, Jena, Germany). Polished thin sections were prepared using these samples.

The study of chromian spinel grains on electron microscopes (Tescan Vega 3sbu and Tescan Vega Compact (Tescan, Brno, Czech Republic)) was conducted at two stages. First, the grains were pasted onto a current-conducting scotch tape using a mushroom-shaped holder and were studied in a secondary electron mode to define their morphological features. Preliminary schemes of the areas were prepared in the "wide field" mode; the grains were numbered, the qualitative assessment of their compositions was provided (chromian spinels, ilmenite, magnetite/hematite, titanomagnetite, titanite, etc.). The exposure mode was as follows: accelerating voltage—20 keV, beam current—30 pA, work distance in the

range of 8–10 mm, scanning mode—"resolution" or "depth". The former proved better for obtaining images of small individual grains (50–100 μm), while the latter was preferable to get images of grain groups or larger grains with quite a deep relief. Prior to the second stage of the research, a thin layer of an epoxy resin was placed onto the samples. Once hardened, the samples were polished.

All possible chromian spinel grains were selected for the morphological study. The grains were divided into six groups (morphogroups) based on their morphological features: (I) regular octahedral crystals with minor defects of vertexes and edges; (II) octahedral crystals with obvious defects of facets, vertexes and edges (flatness, blunting edges, curvature, etc.); (III) octahedron-shaped crystals with extra facets (so-called myriohedral crystals); (IV) crystals of non-octahedral habit (dodecahedra, polyhedra with a trapezohedral, rhombohedral appearance, elongated combinational polyhedra); (V) hard-to-diagnose fragments with a conchoidal fracture or corroded comminuted grains; (VI) whole xenomorphic grains. In total, 464 chromian spinel grains were selected for a detailed morphological analysis. The number of grains selected varied depending on their chromian spinel content and the morphological variety of their grains. Patterns of distribution and incidence of certain morphological groups vary in different placers (Table 1), as explicitly described below (Section 4).

**Table 1.** Composition of morphogroups of chromian spinels in chromite paleoplacers of the Southern Pre-Urals.

| Name of Section | Number of Grains Studied | Morphogroup | | | | | |
|---|---|---|---|---|---|---|---|
| | | 1 | 2 | 3 | 4 | 5 | 6 |
| | | Content, % | Content, % | Content, % | Content, % | Content, % | Content, % |
| Sabantuy | 78 | 23 | 39 | 19 | 19 | 0 | 0 |
| Kolkhoznyi Prud | 82 | 4 | 5 | 18 | 35 | 20 | 18 |
| Verkhne-Yaushevo | 51 | 0 | 4 | 18 | 22 | 20 | 37 |
| Sukhoy Izyak | 36 | 0 | 3 | 19 | 42 | 19 | 17 |
| Bazilevo | 37 | 0 | 8 | 8 | 32 | 16 | 35 |
| Novomikhaylovka | 58 | 5 | 28 | 22 | 25 | 12 | 9 |
| Kiryushkino | 98 | 3 | 16 | 25 | 39 | 12 | 5 |

The second stage of the study involved the quantitative chemical analysis of grains after polishing. Some grains (up to 5%) were lost in polishing. Measurements were conducted in the mode of reflected electrons, at the distance of 15 mm, with the accelerating voltage of 20 keV, beam current of 4 nA and beam diameter of 1–3 μm, at the point mode (spectra were measured from 1,000,000 pulses). The composition was estimated using the AzTec program in an automatic mode with factory standards applied (synthetic and natural compounds).

The chromian spinels formulae were calculated on the basis of four O apfu (atom per formula unit) with fixing of the cation sum $A^{2+} + B^{3+,4+} = 3.00$ and the $Fe^{2+}/Fe^{3+}$ ratio was calculated by charge balance. Indicative geochemical values were estimated using cations $Cr\# = Cr/(Cr + Al)$, $Mg\# = Mg/(Fe^{2+} + Mg)$, $Fe_t = Fe^{2+} + Fe^{3+}$.

## 4. Description of Placers

### 4.1. Sabantuy Paleoplacer

The Sabantuy paleoplacer was discovered first and studied most comprehensively [21,74]. The ore bed has the stable thickness (0.9–1.0 m) and the greatest area (16,500 m², 330 m long and 50 m wide). Estimated resources (P₂ category) of chromite ores with the average $Cr_2O_3$ content of 11 wt.% and the average density of the ore of 3.2 g/cm³ are estimated at 50,160 t (15,675 m³). The chromite-bearing horizon lies at the depth of 0.7–1.5 m (in the southern part) to 3–4 m (in the northern part) from the surface (Figure 2a).

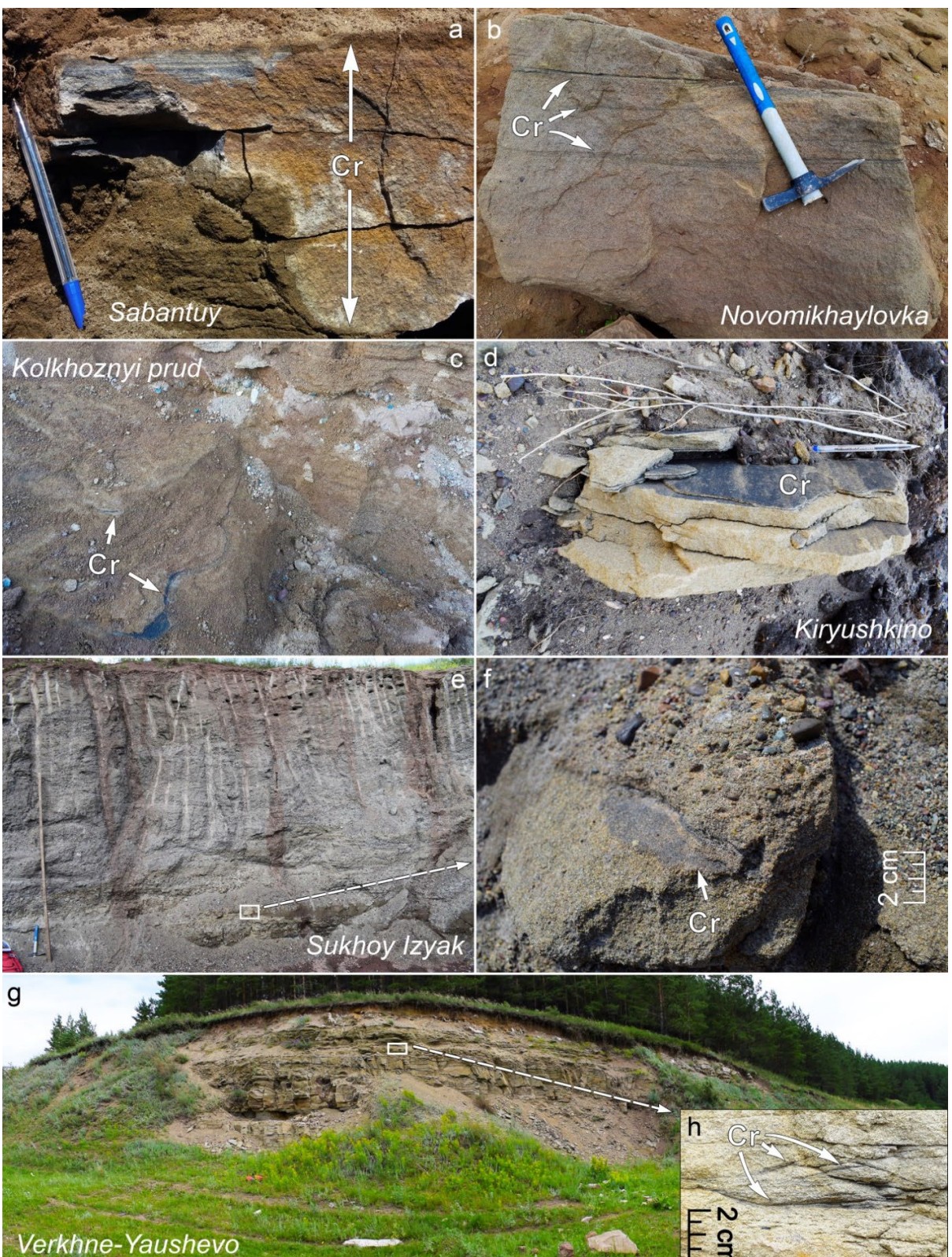

**Figure 2.** An outcrop of chromite-bearing deposits in the Southern Pre-Urals: (**a**)—chromite sandstone in the pit wall, (**b**)—sandstone block with chromite-rich layers, (**c**)—chromite-rich layers in sandy-gravel quarry wall, (**d**)—fragments of chromite sandstones in a quarry, (**e**)—gravel quarry wall with a chromite sandstone lens, (**f**)—enlarged fragment of image (**e**), (**g**)—abandoned quarry wall with chromite sandstones, (**h**)—cross-bedding chromite sand layers (enlarged fragment of image (**g**)).

The studied section with the total thickness of 15 m is described at the left bank of Malaya Berkutla Creek. The ore bed underlies a sequence of light gray recrystallized limestones with the average thickness of 2 m and a sequence of gray subhorizontal fine-grained sandstones with the thickness of 1.1 m (Figure 3). Concentrated chromite sandstones are as thick as 1 to 130 mm and typically show a varied layering, i.e., horizontal, gently undulated, cross-undulated, cross-multidirectional and slanting and sinuous at places. Two former types prevail, but their distribution patterns are still unclear. The chromite bed overlies fine-grained sandstones with subhorizontal and cross bedding that contain interlayers of gravel and pebbles in the upper part of the section. Judging by the structure of the sections 2–3 km west of the Sabantuy paleoplacer, bedded below are gray horizontally layered fine-grained sandstones with fragments of brachiopods and bluish gray siltstones with a total thickness of at least 20 m.

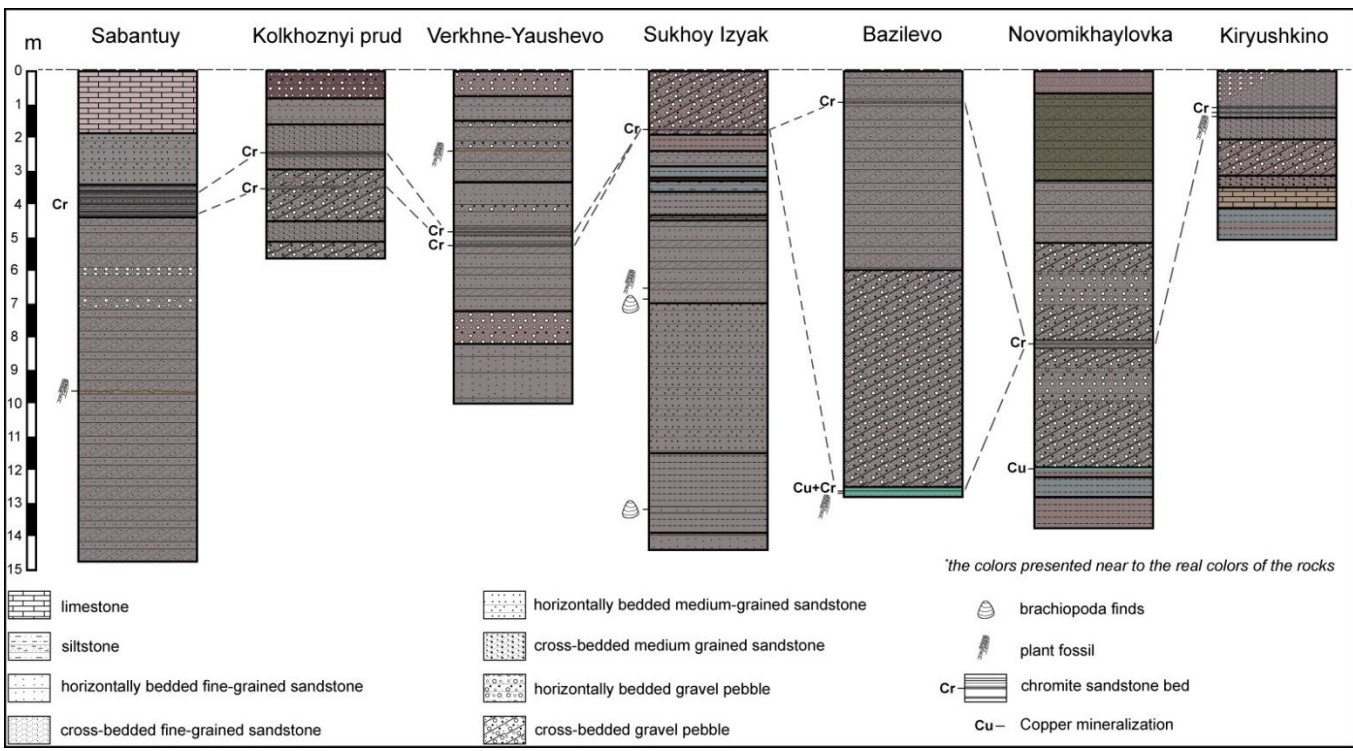

**Figure 3.** Lithology of the studied chromite-bearing deposits in the Southern Pre-Urals.

The sandstone hosting the chromite-rich bed is typically well-sorted and contain well-rounded clasts. The cement of the sandstone is pore-filling carbonate (mainly calcite). The size of particles is 0.3–0.4 mm. Sand particles occur as lithic fragments (77%–82%), quartz (9%–12%), silicate (6%–10%) and ore minerals (1%). The sandstones underlying the ore bed also contain porous-type carbonate cement, but they show a smaller size of particles (0.2 mm), less-rounded and irregularly (well- to medium-) sorted clasts. The sandstones overlying the chromite-rich bed also contain irregularly sorted and rounded clasts, the size of their particles is 0.3–0.4 mm. The studied sandstones fall in the field of litharenites on the ternary QRF diagram (Figure 4a) and in the field of phyllarenites on the SRF-VRF-MRF diagram (Figure 4b). More than a half of the rock clasts are metamorphic (crystalline schists, quartzites, serpentinites) and fewer for sedimentary (siltstones, argillaceous schists) rocks, while detrital magmatic rocks are not more than 3%. The sandstones under- and overlying the chromite-rich bed are compositionally similar to the sandstones interlayering with chromite sandstones. The latter contain a little bit more fragments of magmatic rocks and silicate minerals (plagioclase, amphibole, clinopyroxene). Magmatic rocks are represented by basalts and andesibasalts with a pilotaxitic and an intersertal texture.

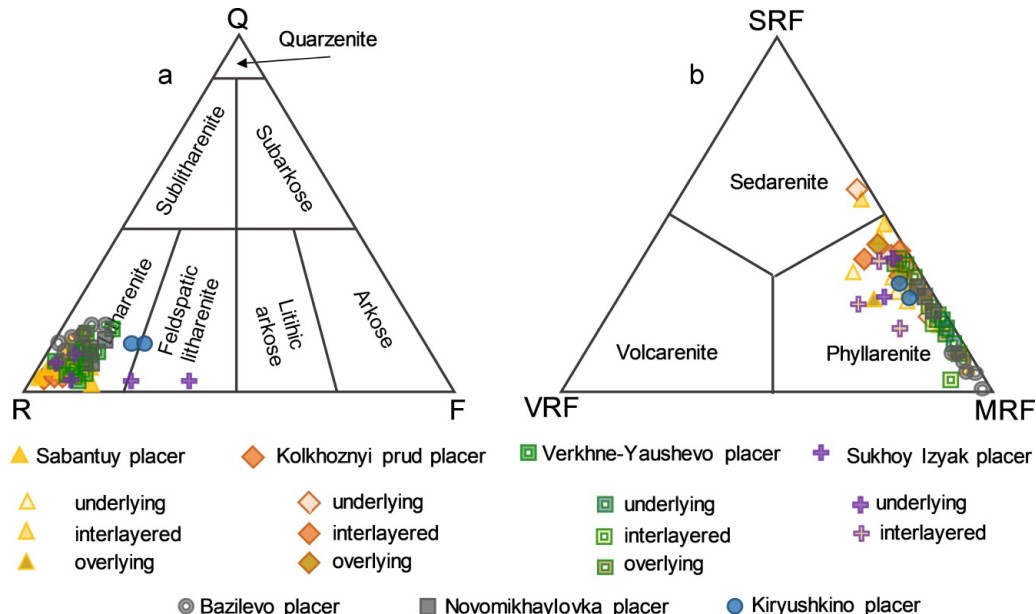

**Figure 4.** Classification ternary diagrams Q–F–R (Quartz–Feldspar–Rock fragments) (**a**) and SRF–VRF–MRF (Sediment rock fragments–Volcanic rock fragments–Metamorphic rock fragments) (**b**) after [76] for the Southern Pre-Urals, chromite-bearing Kazanian-stage sandstones.

Out of four combined samples ($D_5$-13b and $D_5$-18—from the southern part of the chromite-rich bed $F_{19}$-1 and $F_{19}$-3—from the central part), 78 grains were selected for further morphological and chemical analyses. Among these, 18 grains (or 23.1% of the total amount) are referred to group I, 30 grains (38.5%) to group II, 15 grains (19.2%) to each of groups III and IV and 0 to groups V and VI. We preliminarily studied the morphology of chromian spinel grains [21], but quantitative analyses were provided for other grains, disregarding their morphological features. Those studies revealed that at least 10% account for ideal octahedral grains, about 30% of all grains belong to distorted octahedral crystals, while crystals transient from an octahedron to varied polyhedra dominate (up to 50%). Single xenomorphic grains are found here as well. The amount of chromian spinel in the ore mineral fraction was estimated at 70% [74].

The current research established that almost all octahedral crystals of groups I and II have the size of 150–200 μm. Crystals of groups III and IV are larger and have a wider size range, i.e., 150–250 μm. Many of them are corroded, i.e., 7 out of 18 in group I, 17 out of 30 in group II, 12 out of 15 in group III and 8 out of 15 in group IV. Most of the studied grains are weakly rounded, if at all, while only single grains are well-rounded. Regular octahedral crystals commonly show minor defects of vertexes, edges and facets unrelated to corrosion. These are rudimentary vertexes and edges, outgrowths (Figure 5a), multilayered knots, ribbing and pressing-in traces on facets. Crystals of group II with a commonly octahedral habitus are typically flattened (Figure 5b) or have extra facets (100, 001 and 110 in addition to 111). Crystals of group III have a recognizable octahedral appearance with a pronounced decrease in facets or occurrence of additional facets (segments of an octahedron, truncated octahedron). Additional facets are usually those of a rhombic dodecahedron, trapezohedron, tetragontrioctahedron. Facets and edges of an octahedron are often distorted (e.g., stretching along the axes of the 4th or 2nd order). Myriohedral crystals are shaped by a mass of fine grains and curved surfaces, among which facets of an octahedron are lost [77]. Group IV crystals are characterized by the absence of an octahedral appearance probably associated with strong distortions of octahedron, as well as dodecahedra. Elongated but well-defined rhombic dodecahedra are rare [21], grains of combination polyhedra with a poorly recognizable dodecahedral habit are more common. Strong distortions along axes produced rhombohedron-shaped polyhedra (Figure 5d), while sometimes trapezohedral and prismatic.

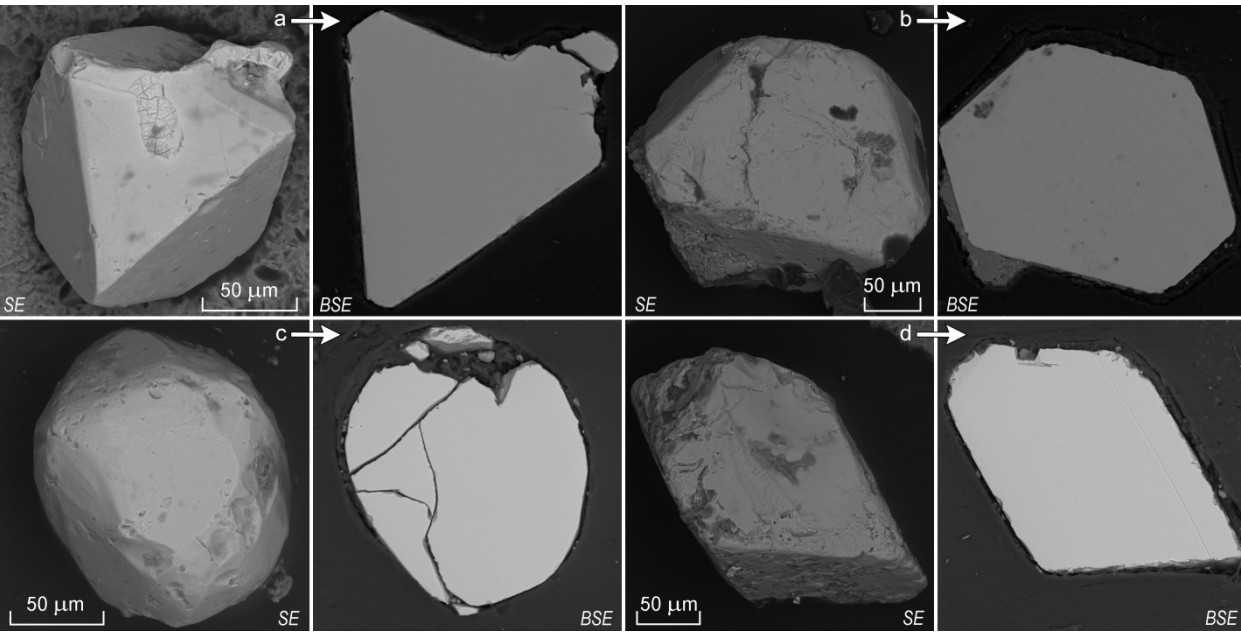

**Figure 5.** Morphology of chromian spinel grains from the Sabantuy paleoplacer (before grinding/after grinding): (**a**)—octahedral crystal with outgrowth (group I), (**b**)—flattened octahedral crystal (group II), (**c**)—myriohedral crystal (group III), (**d**)—rhombohedron-shaped crystal (group IV).

After the grains were ground and polished, some of them were found to have a heterogeneous inner structure, i.e., zonation (patchy, banded, rarely concentric), or various inclusions and fractures, some of them filled in by low-temperature minerals (silicates, calcite, quartz, etc.). However, in general, most grains in the Sabantuy paleoplacer show a solid homogenous structure (~80%). In some crystals primary inclusions were founded. We had formerly distinguished three types of primary inclusions in chromian spinels of the Sabantuy paleoplacer, i.e., monomineral, polymineral and glass inclusions [21]. The glass inclusions were discovered in three grains of group I, i.e., regular octahedral crystals (Figure 6a). Melt inclusions located within octahedral crystals. Amphibole (Figure 6b) and polymineral inclusions (Figure 6c) were also discovered in octahedral crystals. Intergrowths of clinopyroxene, amphibole, plagioclase and chlorite commonly occur in polymineral inclusions. These inclusions are interpreted as recrystallized melt inclusions [21]. At the same time, glass inclusions are interpreted as captured non- or partially recrystallized melt portions. Judging by the shapes of their sections, previously found inclusions of olivine, clinopyroxene and orthopyroxene occur in non-octahedral crystals referred to morphogroups III and IV (Figure 6d).

Chemically, chromian spinels from different morphogroups are closely approximated, except for group I. Chromian spinels with a regular octahedral shape show a markedly high average content of $TiO_2$ (0.49 wt.%), more than twice as high than in chromian spinels of other groups (Table 2). Their $Cr_2O_3$ content varies from 28.8 to 58.7 wt.%. The $Al^{3+}$–$Cr^{3+}$–$Fe^{3+}$ diagram (Figure 7b) demonstrates that compositional points for chromian spinels of morphogroup I are shifted towards the vertex $Fe^{3+}$ compared to chromian spinels of groups II, III and IV corresponding to chromite and alumochromite. Chromian spinels of group II are the highest in Cr ($Cr_2O_3$ up to 69.1 wt.%, Cr# = 0.78) and low in Mg (MgO 9.1 wt.%, Mg# = 0.45). The highest average concentrations of ZnO (0.13 wt.%) were defined in chromian spinels of morphogroups III and IV. Groups III and IV show a greater number of corroded grains compared to groups I and II, while the Zn content in non-corroded grains is commonly below the detection limit of EDS.

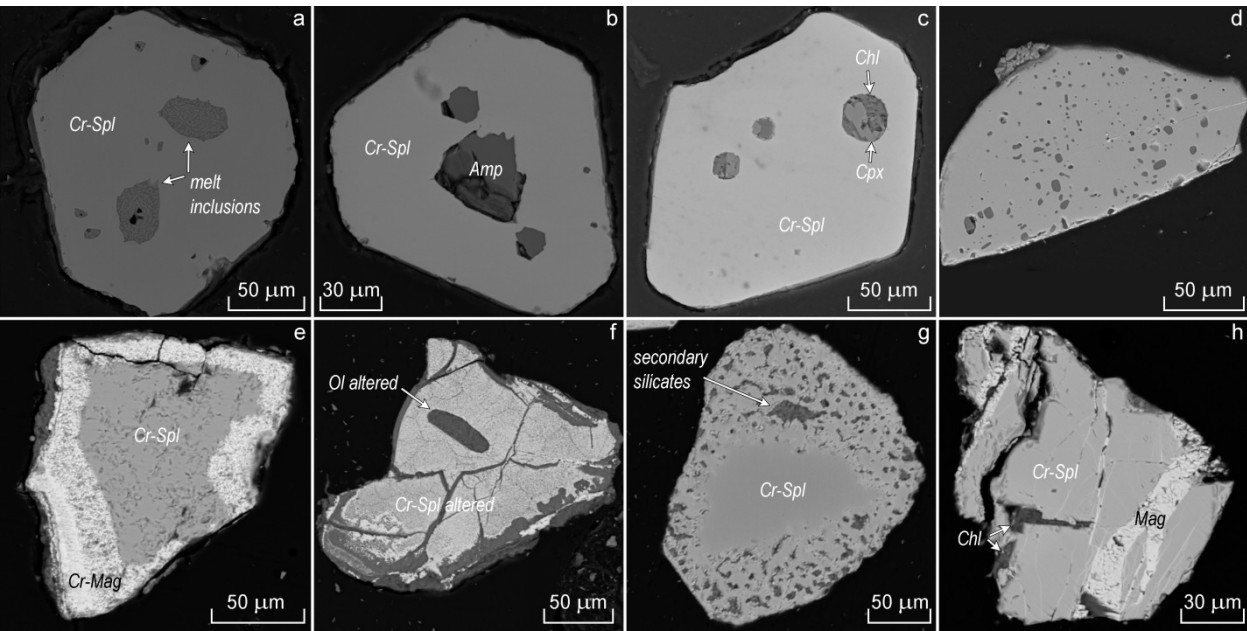

**Figure 6.** BSE-images of chromian spinels from Southern Pre-Ural paleoplacers: (**a**–**d**)—different inclusions in chromian spinels (Sabantuy paleoplacer): (**a**)—melt inclusions, (**b**)—amphibole inclusions, (**c**)—polymineral inclusions, (**d**)—numerous olivine and orthopyroxene inclusions; (**e**)—altered rim of chromian spinel grain (Kolkhoznyi Prud paleoplacer); (**f**)—altered chromian spinel grain with altered olivine inclusion (Sukhoy Izyak paleoplacer); (**g**)—altered rim (spongy texture) and solid core of chromian spinel grain; (**h**)—altered chromian spinel grain with magnetite and chlorite veins (Novomikhaylovka paleoplacer). Amp = amphibole, Chl = chlorite, Cpx = clinopyroxene, Cr-Mag = chrommagnetite, Cr-Spl = chromian spinel, Mag = magnetite, Ol = olivine.

**Table 2.** Average chemical composition of chromian spinels from different morphogroups in all studied placers.

| Placer | Morphogroup | MgO | $Al_2O_3$ | $SiO_2$ | $TiO_2$ | $Cr_2O_3$ | FeO | ZnO | Total | Number of Analyses |
|---|---|---|---|---|---|---|---|---|---|---|
| Sabantuy | I | 9.81 | 13.12 | 0.14 | 0.49 | 48.77 | 26.26 | 0.05 | 98.64 | 15 |
| | s.d. | 3.82 | 7.87 | 0.07 | 0.43 | 8.29 | 7.77 | 0.10 | | |
| | II | 9.13 | 11.19 | 0.10 | 0.18 | 56.45 | 22.36 | 0.07 | 99.47 | 27 |
| | s.d. | 2.72 | 4.56 | 0.10 | 0.15 | 5.27 | 4.27 | 0.19 | | |
| | III | 10.35 | 14.56 | 0.08 | 0.10 | 52.02 | 22.09 | 0.13 | 99.32 | 14 |
| | s.d. | 2.20 | 5.47 | 0.10 | 0.13 | 5.08 | 4.13 | 0.17 | | |
| | IV | 9.89 | 13.43 | 0.11 | 0.13 | 53.96 | 21.78 | 0.13 | 99.43 | 15 |
| | s.d. | 2.58 | 5.22 | 0.08 | 0.13 | 5.36 | 4.17 | 0.19 | | |
| Kolkhoznyi Prud | I | 12.33 | 15.09 | 0.18 | 0.52 | 51.86 | 19.84 | 0.06 | 99.88 | 5 |
| | s.d. | 3.90 | 6.41 | 0.11 | 0.80 | 10.23 | 5.89 | 0.13 | | |
| | II | 13.42 | 5.57 | 0.15 | 0.19 | 57.16 | 21.97 | 0.49 | 98.94 | 5 |
| | s.d. | 2.88 | 5.21 | 0.22 | 0.12 | 4.70 | 9.53 | 0.67 | | |
| | III | 9.65 | 13.63 | 0.16 | 0.24 | 49.56 | 26.05 | 0.06 | 99.35 | 10 |
| | s.d. | 3.12 | 6.15 | 0.23 | 0.21 | 7.15 | 8.96 | 0.14 | | |
| | IV | 8.80 | 13.74 | 0.22 | 0.21 | 48.83 | 27.90 | 0.16 | 99.86 | 32 |
| | s.d. | 3.55 | 9.41 | 0.73 | 0.21 | 11.72 | 13.68 | 0.17 | | |
| | V | 10.65 | 17.27 | 0.03 | 0.08 | 50.93 | 21.20 | 0.12 | 100.28 | 14 |
| | s.d. | 2.81 | 7.23 | 0.11 | 0.18 | 8.16 | 7.71 | 0.17 | | |
| | VI | 8.53 | 15.91 | 0.15 | 0.08 | 41.69 | 32.17 | 0.29 | 98.81 | 19 |

**Table 2.** *Cont.*

| Placer | Morphogroup | MgO | Al₂O₃ | SiO₂ | TiO₂ | Cr₂O₃ | FeO | ZnO | Total | Number of Analyses |
|---|---|---|---|---|---|---|---|---|---|---|
| | s.d. | 4.60 | 13.48 | 0.18 | 0.16 | 16.27 | 23.13 | 0.23 | | |
| | II | 9.99 | 8.67 | 0.12 | 0.16 | 60.10 | 20.97 | 0.00 | 100.01 | 3 |
| | s.d. | 0.70 | 2.90 | 0.11 | 0.14 | 6.95 | 4.61 | 0.00 | | |
| | III | 11.99 | 16.21 | 0.16 | 0.32 | 54.60 | 16.36 | 0.04 | 99.68 | 6 |
| | s.d. | 1.30 | 10.04 | 0.13 | 0.37 | 14.86 | 4.68 | 0.09 | | |
| Verkhne-Yaushevo | IV | 7.55 | 13.98 | 1.01 | 0.37 | 36.13 | 40.12 | 0.11 | 99.28 | 7 |
| | s.d. | 4.28 | 8.18 | 2.38 | 0.40 | 12.41 | 21.40 | 0.20 | | |
| | V | 13.50 | 26.66 | 0.11 | 0.12 | 41.79 | 17.78 | 0.00 | 99.97 | 6 |
| | s.d. | 2.79 | 15.11 | 0.13 | 0.10 | 15.35 | 3.03 | 0.00 | | |
| | VI | 10.39 | 21.55 | 0.08 | 0.16 | 42.88 | 24.53 | 0.04 | 99.65 | 6 |
| | s.d. | 3.61 | 8.71 | 0.14 | 0.13 | 7.88 | 8.45 | 0.10 | | |
| | II | 13.69 | 36.10 | 0.26 | 0.03 | 35.05 | 15.55 | 0.18 | 100.86 | 4 |
| | s.d. | 0.22 | 3.59 | 0.52 | 0.06 | 2.86 | 2.16 | 0.21 | | |
| | III | 10.47 | 18.34 | 0.37 | 0.14 | 48.29 | 21.97 | 0.12 | 99.69 | 11 |
| | s.d. | 3.01 | 6.32 | 1.23 | 0.47 | 7.96 | 13.28 | 0.21 | | |
| Sukhoy Izyak | IV | 7.31 | 13.21 | 1.14 | 0.19 | 52.68 | 24.77 | 0.13 | 99.42 | 39 |
| | s.d. | 4.03 | 8.65 | 1.77 | 0.25 | 12.94 | 12.97 | 0.17 | | |
| | V | 11.31 | 18.64 | 0.25 | 0.12 | 50.73 | 18.27 | 0.09 | 99.41 | 19 |
| | s.d. | 3.46 | 6.90 | 0.57 | 0.32 | 8.29 | 5.42 | 0.16 | | |
| | VI | 7.50 | 14.51 | 1.42 | 0.13 | 48.66 | 26.11 | 0.03 | 98.37 | 8 |
| | s.d. | 3.51 | 6.11 | 2.63 | 0.21 | 7.05 | 10.53 | 0.09 | | |
| | II | 14.20 | 25.90 | 0.00 | 0.21 | 41.48 | 18.29 | 0.04 | 100.11 | 13 |
| | s.d. | 4.31 | 13.87 | n.d. | 0.15 | 13.16 | 5.77 | 0.09 | | |
| | III | 5.85 | 4.20 | 3.27 | 0.00 | 28.53 | 56.51 | 0.40 | 98.77 | 2 |
| | s.d. | 1.49 | 2.91 | 4.63 | n.d. | 11.76 | 1.55 | 0.07 | | 2 |
| Bazilevo | IV | 7.00 | 8.90 | 0.33 | 0.11 | 47.83 | 35.59 | 0.15 | 99.91 | 25 |
| | s.d. | 4.06 | 7.72 | 0.87 | 0.19 | 14.38 | 20.93 | 0.24 | | |
| | V | 10.16 | 18.24 | 0.39 | 0.08 | 41.50 | 29.86 | 0.06 | 100.29 | 10 |
| | s.d. | 4.77 | 13.41 | 0.96 | 0.12 | 14.28 | 23.52 | 0.12 | | |
| | VI | 8.26 | 14.96 | 0.58 | 0.10 | 35.99 | 40.05 | 0.21 | 100.15 | 9 |
| | s.d. | 5.62 | 14.39 | 1.29 | 0.22 | 14.69 | 25.88 | 0.32 | | |
| | I | 8.74 | 11.19 | 0.00 | 0.27 | 57.91 | 21.89 | 0.00 | 100.00 | 1 |
| | s.d. | n.d. | n.d. | n.d. | n.d. | n.d. | n.d. | n.d. | | |
| | II | 8.75 | 12.72 | 0.05 | 0.27 | 52.94 | 25.35 | 0.09 | 100.16 | 22 |
| | s.d. | 2.47 | 4.90 | 0.21 | 0.30 | 6.48 | 3.91 | 0.16 | | |
| Novo-mikhaylovka | III | 10.75 | 14.72 | 0.01 | 0.24 | 52.36 | 22.14 | 0.11 | 100.33 | 21 |
| | s.d. | 2.82 | 8.14 | 0.03 | 0.24 | 9.36 | 4.76 | 0.24 | | |
| | IV | 10.23 | 16.68 | 0.14 | 0.13 | 48.23 | 24.48 | 0.11 | 100.00 | 18 |
| | s.d. | 3.23 | 9.36 | 0.58 | 0.16 | 9.87 | 11.66 | 0.16 | | |
| | V | 11.30 | 17.80 | 0.00 | 0.07 | 50.96 | 20.06 | 0.09 | 100.28 | 10 |
| | s.d. | 2.95 | 9.94 | n.d. | 0.11 | 9.41 | 3.39 | 0.14 | | |
| | VI | 8.56 | 13.95 | 0.03 | 0.03 | 47.45 | 30.32 | 0.13 | 100.45 | 7 |
| | s.d. | 4.05 | 8.44 | 0.08 | 0.07 | 13.88 | 22.09 | 0.16 | | |
| | I | 9.10 | 7.98 | 0.00 | 0.00 | 64.80 | 19.86 | 0.41 | 102.15 | 1 |
| | s.d. | n.d. | n.d. | n.d. | n.d. | n.d. | n.d. | n.d. | | |
| | II | 8.89 | 12.69 | 0.04 | 0.14 | 54.29 | 23.91 | 0.14 | 100.11 | 18 |
| | s.d. | 2.02 | 4.46 | 0.10 | 0.15 | 4.70 | 4.65 | 0.15 | | |
| Kiryushkino | III | 7.93 | 11.17 | 0.12 | 0.10 | 54.78 | 25.49 | 0.15 | 99.73 | 18 |
| | s.d. | 3.29 | 5.56 | 0.27 | 0.16 | 8.32 | 8.26 | 0.30 | | |
| | IV | 9.10 | 13.92 | 0.13 | 0.15 | 52.03 | 24.68 | 0.11 | 100.12 | 36 |
| | s.d. | 2.35 | 5.66 | 0.48 | 0.13 | 7.96 | 9.22 | 0.17 | | |
| | V | 11.96 | 16.52 | 0.00 | 0.12 | 53.73 | 18.28 | 0.04 | 100.65 | 12 |
| | s.d. | 2.57 | 8.32 | 0.00 | 0.13 | 8.09 | 3.81 | 0.10 | | |
| | VI | 8.84 | 8.65 | 0.09 | 0.00 | 54.26 | 28.82 | 0.00 | 100.65 | 4 |
| | s.d. | 4.77 | 5.45 | 0.18 | 0.00 | 12.86 | 20.08 | 0.00 | | |

Note: "s.d."—standard deviation, "n.d."—no data.

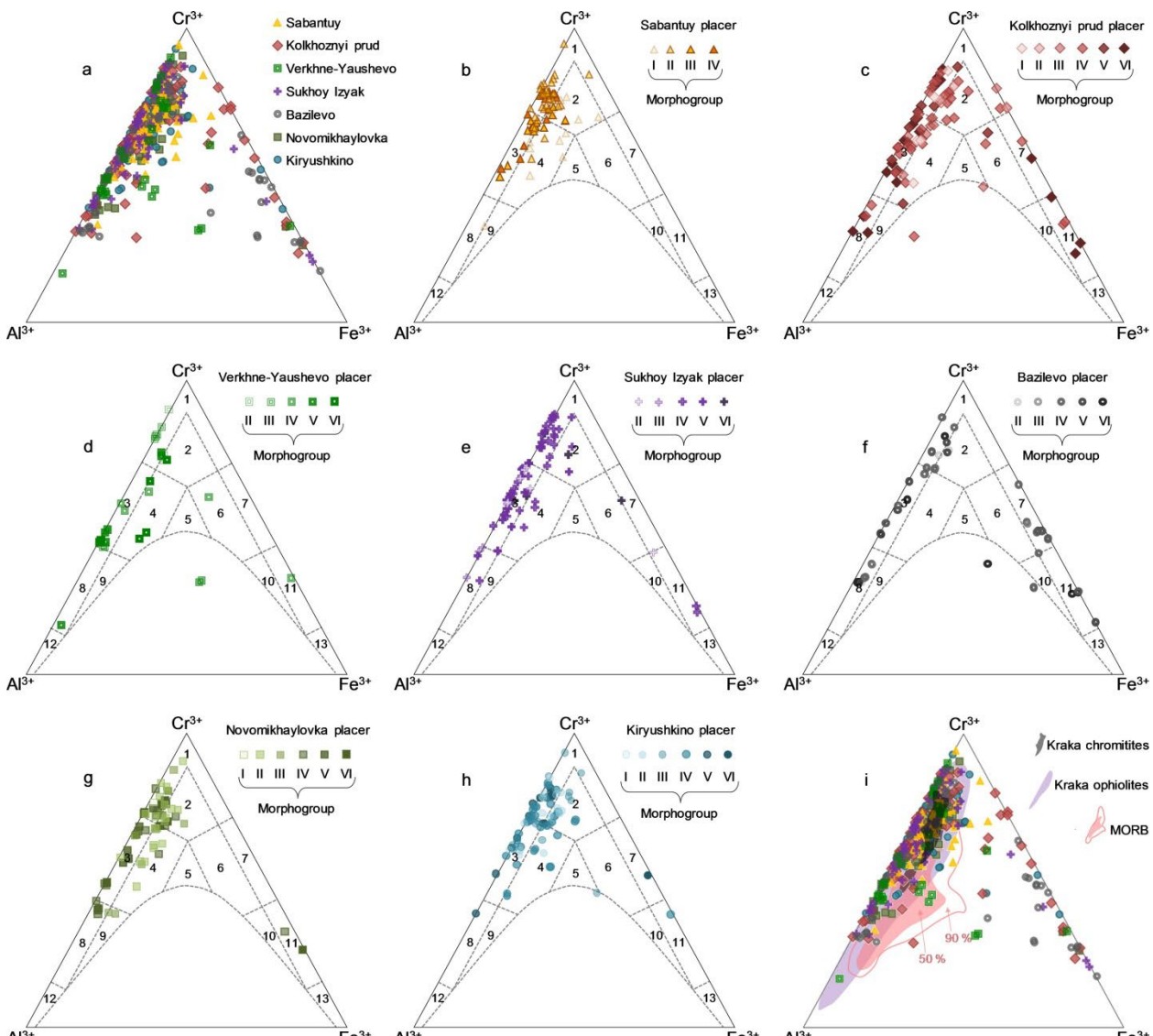

**Figure 7.** Ternary $Al^{3+}$–$Cr^{3+}$–$Fe^{3+}$ diagrams for chromian spinels from Southern Pre-Urals chromite paleoplacers: (**a**)—compositions of all placers together, (**b**)—compositions of Sabantuy placer, (**c**)—compositions of Kolkhoznyi Prud placer, (**d**)—compositions of Verkhne-Yaushevo placer, (**e**)—compositions of Sukhoy Izyak placer, (**f**)—compositions of Bazilevo placer, (**g**)—compositions of Novomikhaylovka placer, (**h**)—compositions of Kiryushkino placer, (**i**)—compositions of both all studied placers, Kraka chromitites and ophiolites [72], and MORB [60]. Fields of chromian spinel compositions are after [78]: 1—chromite, 2—subferrichromite, 3—alumochromite, 4—subferrialumochromite, 5—ferrialumochromite, 6—subalumoferrichromite, 7—ferrichromite, 8—chrompicotite, 9—subferrichrompicotite, 10—subalumochrommagnetite, 11—chrommagnetite, 12—picotite, 13—magnetite.

Notably, we had formerly established that bulk compositions of chromian spinels of different samples from the lower, upper, southern and central parts of the chromite-rich bed are similar [21]. Full analytical data on chromian spinels are provided in Supplementary Table S1.

### 4.2. Kolkhoznyi Prud Paleoplacer

The Kolkhoznyi Prud paleoplacer was discovered in a gravel sandy quarry 4.7 km east of the Sabantuy paleoplacer (Figure 1b). The area of chromite placer is ≥500 m². The studied section (Figure 3) with the total thickness of ~6 m is represented by the

following types of sediments: brownish red subhorizontal gravel pebbles (0.55 m), gray subhorizontal fine-grained sandstones (0.52 m), gray medium- to coarse-grained cross-bedded sandstones (0.9 m), brownish gray cross-bedded gravel pebbles with sandstone lenses (1.04 m), brownish gray cross-bedded fine- to medium-grained sandstones (0.41 m) and brownish gray cross-bedded gravel pebbles (2.35 m).

Chromite sandstones form the upper subhorizon in interlayers of subhorizontally bedded sandstones and the lower subhorizon in cross-bedded sandstones and in thin lenses within gravel-pebble sediments. In the upper ~30 cm-thick subhorizon chromite sandstones produce thin (1–3 mm) interlayers (Figure 2c), 12–15 in total. The lower 10–15 cm-thick horizon contains the same interlayers, 8–10 in total. The sandstones enclosing chromite-rich interlayers show well-sorted and poorly-rounded grains. The cement is porous, compositionally carbonate. The size of particles is 0.3 mm, and they occur as lithic fragments. Sandstones underlying the chromite-rich bed differ only in their better-rounded clasts and a slightly larger size of its particles (0.4 mm). Sandstones from the overlying sequence have the clayey-ferruginous cement. Clasts of these sandstones are compositionally similar to rocks of the Sabantuy paleoplacer. Rock debris dominates (77%–86%), where 50% account for metamorphic rocks. The classification ternary diagrams show that figurative points overlap with compositions of sandstones from other placers (Figure 4).

The chromian spinel content in the heavy mineral fraction of sample $B_{21}$-64 is ~40%. Out of 82 separated chromian spinel grains, only 3 (3.7%) belong to group I, 4 grains (4.9%) to group II, 15 grains (18.3%) to group III, 29 grains (35.4%) to group IV, 16 grains (19.5%) to group V and 15 grains (18.3%) to group VI. Grains of groups I and II are in no way different from grains of the Sabantuy paleoplacer. Group II contains truncated and curved octahedron. All of them have a corroded surface. Grains of groups III and IV, unlike those in the Sabantuy paleoplacer, show far worse-shaped crystals and are mainly subidiomorphic. Myriohedral grains (rounded rudimentary dodecahedra) and tritetrahedron-shaped grains are widespread among crystals of group III (Figure 8a). Rudimentary or distorted (e.g., elongated or containing additional facets) rhombododecahedron (Figure 8b) and trapezohedron-shaped crystals with poorly formed facets are abundant in group IV. Grains of group V are mainly represented by debris with a conchoidal surface with no traces of rounding or corrosion along the fracture (Figure 8c). According to the outlines of some grains, underdeveloped dodecahedra and crystals of a trapezohedral and tritetrahedral shape are assumed. Xenomorphic grains of group VI are commonly slightly flattened (Figure 8d) and rarely isometric. They usually have no flat surfaces, but show numerous knots or cavities. The Kolkhoznyi Prud paleoplacer does not contain well-rounded chromian spinel grains, and weakly rounded grains are minor. Grains of groups I, II and III have the size of 100–150 μm, while grains of groups IV, V and VI are larger, with their size ranging from 180 to 250 μm.

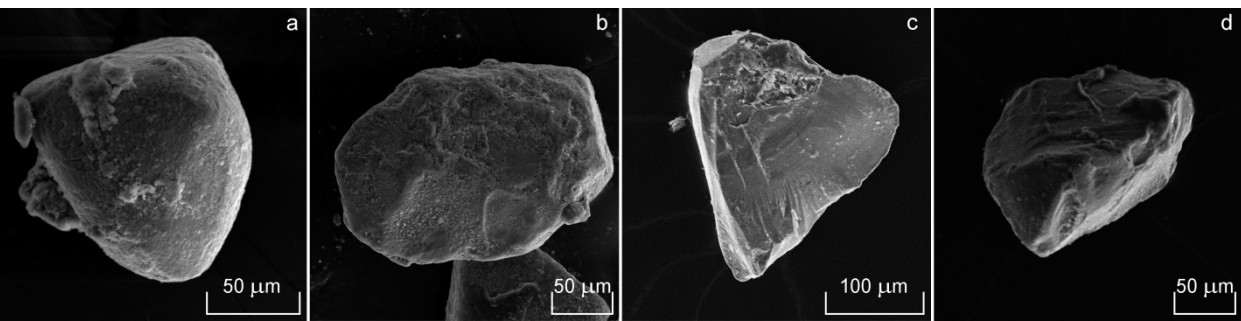

**Figure 8.** Morphology of chromian spinel grains from the Kolkhoznyi Prud paleoplacer (SE images): (**a**)—segment of the octahedron (tritetrahedron-shaped crystal) (group III), (**b**)—distorted prolate dodecahedron (group IV), (**c**)—crystal fragment (group V), (**d**)—xenomorphic grain (group VI).

Internally, the grains showed that only a minor part (~25%) had a dense homogenous structure, while most of them were porous (spongeous). Many grains show the heteroge-

nous inner structure associated with irregular distribution of magnetitization zones, i.e., along the rim (Figure 6e), as spots or strips. Along cleavage cracks chromian spinels are often substituted by not only magnetite, but also by low-temperature silicates (e.g., chlorite and hydromicas).

The chemical composition of chromian spinels from various morphogroups has some differences (Table 2). Chromian spinels from group VI show the most varied contents of $Cr_2O_3$ (16–65 wt.%, Cr# = 0.31–0.98) and other elements. On $Al^{3+}$–$Cr^{3+}$–$Fe^{3+}$ diagram these chromian spinels correspond to chromite, alumochromite, chrompicotite and chrommagnetite (Figure 7c). The highest average $TiO_2$ content (0.52 wt.%) is typical of chromian spinels of group I, just like in the Sabatuy paleoplacer. Chromian spinels of group II are the highest in Cr (Cr# = 0.89) and high in Mg (Mg# = 0.66). Chromian spinels of group V are the lowest in Cr (Cr# = 0.67), while their average $Al_2O_3$ content is the highest (17.3 wt.%). High values of Zn (up to 1.3 wt.%) are recorded in corroded grains of chromian spinels. In general, most figurative points fall in compositional fields of chromite and alumochromite (Figure 7c).

*4.3. Verkhne-Yaushevo Paleoplacer*

The Verkhne-Yaushevo paleoplacer was discovered in a wall of an abandoned sand quarry 7.1 km east of the Sabantuy paleoplacer (Figures 1b and 2g). The placer area is unknown. Brownish red subhorizontal gravel pebbles with the thickness of 0.5 m are exposed in the upper part of the section (Figure 3). Subhorizontally bedded below is a 0.5 m-thick sequence of gray fine-grained sandstones that overlap cross-bedded sandstones with interlayers of gravel and pebbles 1.24 m thick in total. The upper and middle parts of the last sequences include horizons of paleosols represented by loose and greasy to the touch brown earthy material. They overlie a 2.6 m-thick sequence of gray subhorizontal fine- to medium-grained sandstones with interlayers of gravel pebbles and cross-bedded sandstones. Numerous 1–3 mm-thick interlayers of chromite sandstones producing a 0.4 m horizon were found in the middle of the sequence. Chromite sandstones are associated with borders of layers of subhorizontally and cross-bedded sandstones. Consequently, the bedding of chromite-rich layers changes from horizontal to angularly slanting (Figure 2h). Polymictic sandstones with chromite-rich layers contain well-sorted and poorly rounded clasts. The cement is clayey and ferruginous-clayey of the closed porous type; residual carbonate cement is observed at places. The average size of particles is 0.35 mm. Sandstones from an overlying chromite-rich bed differ only in irregularly sorted clasts expressed as a layering with the typical size of grains is 0.2–0.3–0.4 mm. The sorting of sandstones underlying the ore horizon is irregular, mainly medium; the carbonate cement is better preserved. Rock clasts (65%–73%) dominates in the composition of sandstones like in the previous placers; the quartz content is more varied (3%–15%) and the amount of silicate minerals is higher (up to 15%). On classification diagrams, compositional points for sandstones from the Verkhne-Yaushevo paleoplacer overlap with sandstones from other paleoplacers (Figure 4).

In sample $B_{21}$-110 from the lower part of the chromite horizon, the chromian spinel content in the heavy mineral fraction is about 90%. Well-facetted crystals are rare, even crystals with distinct crystallographic forms have rudimentary facets and vertexes. Out of 51 extracted chromian spinel grains none belongs to group I, 2 grains (3.9%) belong to group II, 9 grains (17.6%) to group III, 11 grains (21.6%) to group IV, 10 grains (19.6%) to group V and 19 grains (37.3%) to group VI. Many grains except fragmental (group V) have a corroded surface. Rounded grains are rare, while un- or weakly rounded grains are the most common. Most grains are referred to the size range of 130–180 μm. Most crystals of group III are octahedron with facets of a dodecahedron (Figure 9a), grains with the appearance of a highly distorted truncated octahedron (Figure 9b). Grains of group IV can be hardly distinguished morphologically. They are represented by crystals with a rudimentary dodecahedron or rhombohedron shape. Only facets of grains from group V are corroded; their fractures were not subject to corrosion. They contain rare

grains with dodecahedron contours (Figure 9c). Grains of group VI are often flattened and slightly elongated (Figure 9d). Grains of this group are the most corroded. Structurally, both homogenous and heterogenous grains are observed. The former are typical of non-corroded crystals, while the latter have a spongeous structure and show corroded crystals. The former account for only ~20% of the total amount.

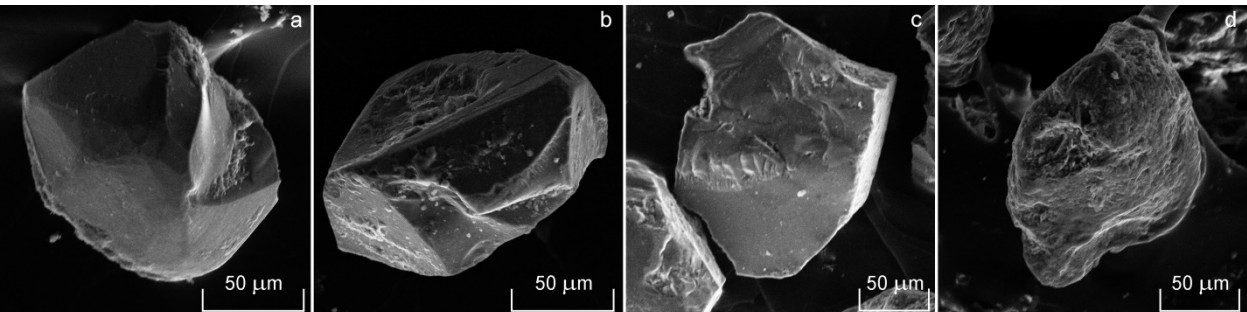

**Figure 9.** Morphology of chromian spinel grains from the Verkhne-Yaushevo paleoplacer (SE images): (**a**)—slightly flattened octahedron with dodecahedron facets (group III), (**b**)—fragment of truncated octahedral bipyramidal crystal (group IV), (**c**)—fragment of crystal (group V), (**d**)—xenomorphic corroded grain (group VI).

The chemical composition of chromian spinels from various morphogroups differs (Table 2). Chromian spinels of group II are the highest in Cr ($Cr_2O_3$ 60.1 wt.% on average, Cr# = 0.88), poor in $Al_2O_3$ (8.7 wt.% on average) and low in Mg (MgO 10.0 wt.%, Mg# = 0.49). Chromian spinels of group IV ($Cr_2O_3$ 36.1 wt.% on average, Cr# = 0.66) and group V ($Cr_2O_3$ 41.8 wt.% on average, Cr# = 0.53) are the lowest in Cr. Noteworthy, all grains in group IV are corroded, while those in group V are not corroded. In addition, chromian spinels of group IV are typically the lowest in Mg (MgO 7.6 wt.%, Mg# = 0.37) and the highest in $TiO_2$ (0.37 wt.% on average) and ZnO (0.11 wt.%). Chromian spinels of group V are the richest in $Al_2O_3$ (26.7 wt.% on average). Most points on $Al^{3+}$–$Cr^{3+}$–$Fe^{3+}$ diagram (Figure 7d) fall in fields of chromite-alumochromite and subferrialumochromite. Compositions of chromian spinels from group II are distributed in a compact way, while points in other groups (especially in group IV) are scattered along the diagram.

### 4.4. Sukhoy Izyak Paleoplacer

The Sukhoy Izyak paleoplacer was discovered in a sandy-gravel quarry 12.2 km west of the Sabantuy paleoplacer (Figure 1b). The observed section starts in the quarry and continues down the hill slope. It is composed of the following rock types: brownish red cross-bedded gravel pebbles (1.3 m), dark red mudstones (0.3 m), gray horizontally bedded fine-grained sandstones (0.3 m), bluish gray siltstones with a 8 cm-thick interlayer of dark brown mudstones (0.5 m), gray siltstones (0.5 m), gray tabular limestones (0.1 m), gray fine-grained cross-bedded sandstones with interlayers of horizontally bedded (1.7 m) sandstones, gray medium-grained horizontally bedded sandstones (3 m), gray siltstones with interlayers of fine-grained sandstones (1.6 m) and gray fine-grained sandstones (not less than 0.4 m) at the end of the section.

The concentrated chromite-rich bed with the thickness of 1–3 cm was discovered in a 10–15 cm-thick convoluted layer of sandstones in the uppermost part of the gravel-pebble unit. Individual chromian spinel interlayers as thick as 1–2 mm are united into a single 10–12 mm layer (Figure 2f) at the placer. The area of the chromite-rich bed was estimated at ≥1100 $m^2$. The chromite-bearing horizon is attributed to the border of coarse-grained (average size—1 mm) and medium-grained (average size—0.4 mm) sandstones. Upper coarse-grained sandstones typically contain opal cement, while lower coarse-grained sandstones have clayey cement, locally relict and carbonate. Lithic fragments dominate in the composition of the sandstone, but their amount is more variable compared to other chromite-bearing sections (46%–79%). In sandstones of the Sukhoy Izyak paleoplacer,

the highest portion of magmatic rocks was defined (6 to 13% out of all rock clasts) and a significant amount of silicate minerals (up to 25%). On the QRF diagram, sandstone compositions are displaced towards feldspatic litharenite, while on the SRF–VRF–MRF diagram, they are visibly displaced towards the VRF vertex compared to the bulk figurative points (Figure 4). The sequences immediately underlying and overlapping the chromite-bearing horizon are represented by friable rocks. No detailed petrographic study was given to them.

In total, 52 grains of ore minerals were extracted from sample $B_{21}$-142, where 36 grains (66%) are chromian spinels and the others are magnetite, hematite, titanomagnetite, titanite and ilmenite. According to morphological studies, no single chromian spinel grain corresponds to group I, only 1 grain (2.8%) belongs to group II, 7 grains (19.4%) to group III, 15 grains (41.7%) to group IV, 7 grains (19.4%) to group V and 6 grains to group VI (16.7%). Many grains are weakly or highly corroded (except for grains of group V with a non-corroded fracture surface). Most grains are weakly rounded, if at all. The size of the grains is 100–250 μm, but the majority (about 80%) is 150–200 μm in size. Grains of group III are represented by weakly formed crystals transient from an octahedron to a dodecahedron or trioctahedron. Grains of group IV are mainly elongated (columnar), and they encompass hard-to-define distorted and rhombohedron-shaped crystals (Figure 10a,b). Grains of group V represented by fragmental crystals, mainly or totally non-octahedral, have elongated forms and sharp angles (Figure 10c). Grains of group VI mainly contain highly corroded, flattened and isometric individuals (Figure 10d). Since many grains are corroded, they have a spongeous structure and magnetitization zones represented by mottles and bands. Silicate minerals occur in pores and cracks. An inclusion of serpentinized olivine was found in one of altered xenomorphic chromian spinel grains (Figure 6f). Grains with a solid homogenous inner structure are rare (~10%).

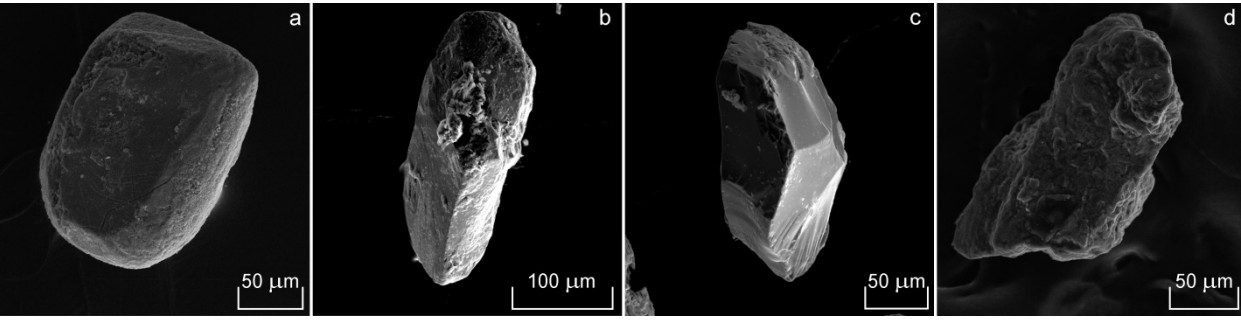

**Figure 10.** Morphology of chromian spinel grains from the Sukhoy Izyak paleoplacer (SE images): (**a**)—flat corroded rhombohedron-like grain (truncated octahedral bipyramidal crystal) (group IV), (**b**)—elongated rhombohedron-shaped grain (group IV), (**c**)—fragment of crystal (group V), (**d**)—xenomorphic corroded grain (group VI).

Most chromian spinels from the Sukhoy Izyak paleoplacer, just like those from other placers, compositionally correspond to chromite and alumocromite (Figure 7e). The selection for group II is not representative and includes only four analyses of the same corroded grain of Ti-low alumochromite. Among other morphogroups, the following pairs with similar chemical compositions were established: groups III and V and groups IV and VI (Table 2). Chromian spinels of groups III and V are high in $Al_2O_3$ (18.3%—III, 18.6%—V), higher in Mg (MgO 10.5 wt.%, Mg# = 0.49 and MgO 11.3 wt.%, Mg# = 0.53 in groups III and V, respectively) and low in Cr ($Cr_2O_3$ 48.3 wt.%, Cr# = 0.65 and $Cr_2O_3$ 50.7, Cr# = 0.65 in groups III and V) compared to chromian spinels of groups IV and VI ($Al_2O_3$ 13.2%—IV, 13.9%—VI, MgO 7.3 wt.% and Mg# = 0.35 both in groups IV and VI, $Cr_2O_3$ 52.7 wt.%, Cr# = 0.74 and $Cr_2O_3$ 47.9 wt.%, Cr# = 0.71, respectively) (Table 2). Chromian spinels of groups III and IV show high concentrations of ZnO (0.12 and 0.13%, respectively), but in chromian spinels of group VI the concentration of Zn exceeds the detection limit only in one sample, though all of these grains are corroded. A xenomorphic grain of titanochromite

was referred to group VI (see in EM), which was discovered in a single grain in the studied sample. It was excluded from calculations of the average composition of the morphogroup. Titanochromite is quite a rare mineral that occurs in some ultrabasites and their weathering products [40,79] and in various Moon rocks [80].

*4.5. Bazilevo Paleoplacer*

The Bazilevo paleoplacer was discovered in a gravel-pebble quarry 13.3 km southwest of the Sabantuy paleoplacer (Figure 1b). Two horizons of stockwork copper ore bodies with the average Cu content of 7.2 wt.% were stripped in this quarry [81]. In addition, two horizons of chromite sandstones were discovered here: (1) the upper horizon (above the copper horizon) with three 1–3 mm-thick interlayers and (2) the lower horizon (within the copper horizon) with several ~0.5 mm-thick interlayers. The area of the chromite placer is $\geq$500 m$^2$. The section is composed of (from top to bottom) subhorizontal brownish-gray fine-grained sandstones (4 m); cross-bedded brownish-gray sandstones intercalated by gravel, pebbles and coarse-grained sandstones (4.5 m). Copper ore bodies are bedded at the bottom of the quarry. It was discovered that the copper mineralization in one of them was superimposed on chromite sandstones, which resulted in formation of malachite-azurite cement in the clastic mass of ore minerals (chromite, magnetite, ilmenite) [81]. This find is very curious, since it allows considering the studied rocks as samples of unusual sedimentary Cu-Cr ores formed at two stages, i.e., (1) sedimentary and (2) fluid-related [81].

The upper chromite-bearing horizon comprises weakly cemented sandstones with the carbonate matrix. They are well-sorted; the composition of clasts was not studied. Sandstones of the lower chromite-bearing horizon have medium- to well-sorted and medium-rounded fragments. The cement is contact-porous, its composition irregularly changes from clayey to siliceous. Relics of carbonate cement are locally observed. The average size of grains is 0.3 mm. The portion of rock clasts ranges from 61 to 85%, where 88%–99% account for fragments of metamorphic rocks, 1%–12% fragments of sedimentary rocks and up to 2% fragments of magmatic rocks. The portion of quartz clasts varies from 10 to 17%, while silicate minerals (feldspar, amphibole, chlorite, pyroxene)—2%–12%. On the SRF-VRF-MRF diagram compositional points for sandstones from the Bazilevo placer are displaced to the MRF top closer than the others (Figure 4).

Sample B$_{21}$-37 from the upper horizon of chromite sandstones was selected for detailed study of chromian spinel grains. Out of 105 separated grains of the heavy mineral fraction only 38 (36%) proved chromian spinel, which is the lowest value for all of the studied placers. Out of these grains, none belongs to group I, 3 grains (8.1%) each in groups II and III, 12 grains (32.4%) in group IV, 6 grains (16.2%) in group V and 13 grains (35.1%) in group VI. The size of their grains is 100–200 μm. The grains are poorly rounded, if at all. Grains of group II are truncated octahedron with minor additional cube facets (Figure 11a). Grains of group III are commonly myriohedral, often rudimentary (Figure 11b). Group IV contains strongly distorted prolate octahedron and dodecahedra with poorly formed facets in the form of rhombohedron-shaped grains (Figure 11c). Grains of group V do not distinguish from the previous placers. Most grains of group VI are also corroded up to the loose surface with numerous plates (Figure 11d). Like in the previous placers (except for the Sabantuy), many chromian spinel grains show a penetrative spongy structure, i.e., pores occur all over the grain area, including the core. The minor part of non- or poorly corroded grains (~35%) has a homogenous structure or a spongy rim with the solid core.

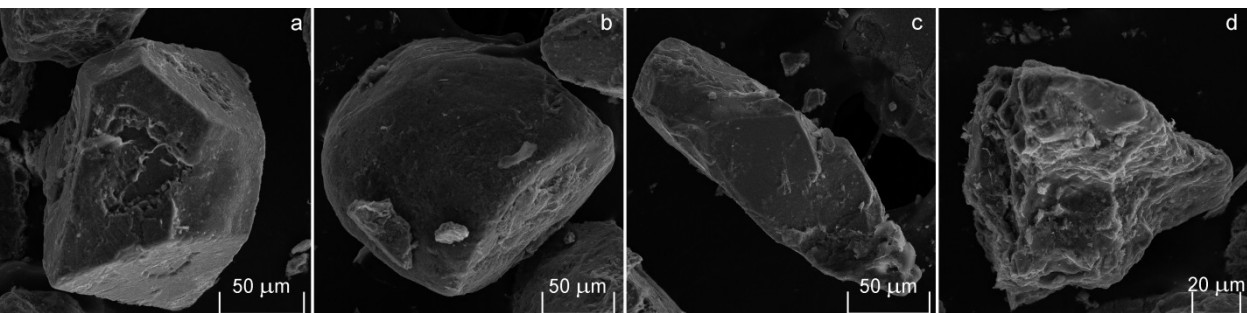

**Figure 11.** Morphology of chromian spinel grains from the Bazilevo paleoplacer (SE images): (**a**)—truncated octahedron (group II), (**b**)—segment of myriohedral grain (group III), (**c**)—defective columnar rhombohedron-shaped grain (group IV), (**d**)—xenomorphic highly corroded grain (group VI).

The chemical composition of chromian spinels from various morphotypes is significantly different, which is partly explained by the representativeness of the selection (Table 2). Thus, group III is only represented by two analyses from two corroded grains of ferrochromite. Unlike other placers, where compositions of chromian spinels are mainly distributed along the $Al^{3+}$–$Cr^{3+}$ facet on the ternary diagram, almost 30% of figurative points are plotted along the $Cr^{3+}$–$Fe^{3+}$ facet here (Figure 7f). Group IV ($Cr_2O_3$ 18.9–71.7 wt.%, Cr# = 0.33–0.99) and group V ($Cr_2O_3$ 11.9–58.4 wt.%, Cr# = 0.31–1.00) show the widest ranges of the Cr content, while chromian spinels in group V are much richer in $Al_2O_3$ on average (18.2 vs. 8.9 wt.%). Chromian spinels of group II are the highest in Mg (MgO 14.2 wt.% and Mg# = 0.62 on average) and $Al_2O_3$-rich (25.9 wt.%), while lowest in Cr ($Cr_2O_3$ 41.5 wt.%, Cr# = 0.54) among all morphogroups. It distinguishes them from chromian spinels of group II of other placers that show a representative chromian spinel selection with a high Cr content (Cr# $\geq$ 0.74). First, this is due to the fact that several grains of group II conform to chromous spinel. Second, all studied grains of group II in the Bazilevo placer are highly corroded, though the ZnO content is low in contrast to chromian spinels of group IV (ZnO 0.15 wt.%) and group VI (ZnO 0.21 wt.%) that are also mainly corroded.

*4.6. Novomikhaylovka Paleoplacer*

The Novomikhaylovka paleoplacer was discovered in a sandy gravel quarry 17.2 km south of the Sabantuy paleoplacer (Figure 1b). The chromite placer area is $\geq$800 m$^2$. The upper part of the section (Figure 3) is composed of 0.44 m-thick brownish subhorizontally bedded fine-grained sandstone. Bedded below are yellowish brown fine-grained cross-bedded sandstones with horizontally bedded interlayers (1.76 m) overlain by the same brownish gray rocks (1.25 m). Below, there are brownish-gray gravel pebbles with sandstone lenses and interlayers of horizontally bedded gravel pebbles with the total thickness of 4.52 m. Right in the middle of this sequence, fine (1–3 mm) interlayers of chromite sandstones were discovered in sandstone interlayers, producing a 10–20 cm ore horizon (Figure 2b). The section ends up with dark red siltstones ($\geq$1.2 m) with a 0.4 m-thick interlayer of bluish gray siltstones. A 3 cm-thick Cu-mineralized layer (chrysocolla, malachite) was discovered in siltstones at the border with gravel pebbles. Sandstones hosting chromite-rich interlayers typically have well-sorted and medium-rounded grains. The cement is carbonate, closed porous. The average size of particles is 0.2 mm. The portion of rock clasts in the detrital fraction of sandstones is 62–64%, where metamorphic rocks account for 64%–69%. The amount of quartz is 6%–9%, silicate minerals are 12%–14%, the portion of ore minerals is high as 5–7%. On classification diagrams, compositional points of these sandstones overlap with the others (Figure 4). Over- and underlying sediments are poorly cemented; their petrography has not been studied.

In the ore mineral fraction of sample $B_{21}$-232, 63% of chromian spinel was defined. Out of 65 separated grains, 3 (4.6%) belong to group I, 18 grains (27.7%) to group II,

14 grains (21.5%) to group III, 16 grains (24.6%) to group IV, 8 grains (12.3%) to group V and 6 grains (9.2%) to group VI. The size of all grains is 150–250 µm, while grains in the range of 150–200 µm are the most common (grains of groups I–III). Grains of group IV fall in the size of 200–250 µm, just like those of the Sabantuy paleoplacer. The grains are un- or poorly rounded as a rule, though well-rounded individuals are also found. Both non-corroded and highly corroded grains occur among octahedral crystals of group I (Figure 12a). Crystals with vicinals (minor facets 110, 011) and hexahedron facets (100, 010) are commonly observed among distorted octahedron of group II. Group III typically contains crystals transient from an octahedron to a dodecahedron and tritetrahedron (Figure 12b) or tetragonal pyramid, as well as distorted crystals of a cuboctedral appearance, complicated by the facets of a dodecahedron (Figure 12c). The crystals are non- or poorly corroded. Group IV often shows elongated strongly distorted dodecahedron and octahedron (Figure 12d) up to the development of rhombohedron- and tritetrahedron-shaped crystals with pyramidal facets. Both weakly and highly corroded grains are found among them. Judging by contours of grains in group V, they are non-octahedral crystals with cleavaged facets and vertexes. The Novomikhaylovka paleoplacer shows a considerably higher percentage of grains with a dense homogenous structure or at least with the solid core (~50%) compared to the previous four placers. Grains with an unaltered core and a porous rim with various silicate minerals are common (Figure 6g). However, there are also grains with penetrative spongy structure and grains with mottled or veined magnetite (Figure 6h).

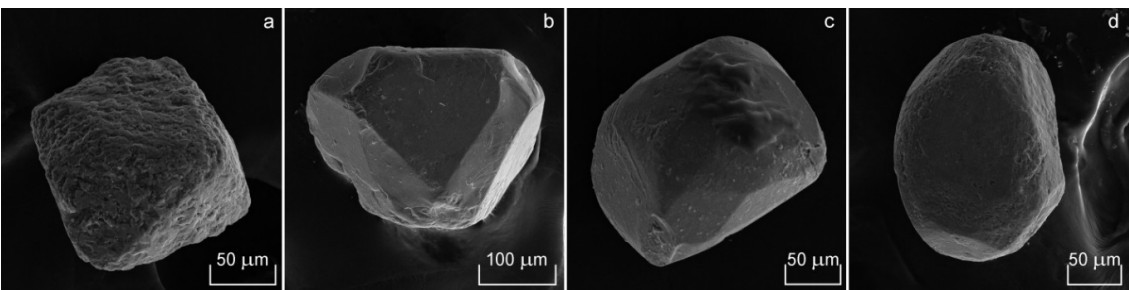

**Figure 12.** Morphology of chromian spinel grains from the Novomikhaylovka paleoplacer (SE images): (**a**)—corroded octahedron (group I), (**b**)—segment of the octahedron (tritetrahedron-shaped crystal with vicinals) (group III), (**c**)—distorted cuboctahedron-shaped crystal complicated by dodecahedron facets (group III), (**d**)—distorted octahedral rhombohedron-shaped crystal with pyramidal facets (group IV).

Chromian spinels of various morphogroups are approximately close in terms of their chemical composition (Table 2), just like in the Sabantuy paleoplacer. The selection for group I is not representative and includes only one analysis of Ti-low chromite. Most composition points on the $Al^{3+}$–$Cr^{3+}$–$Fe^{3+}$ diagrams fall in fields of chromite, while fewer in fields of subferrichromite and alumochromite (Figure 7g). Chromian spinels of group II are the highest in Cr ($Cr_2O_3$ 52.9 wt.% on average, Cr# = 0.74) and the lowest in Mg (Mg 8.7 wt.%, Mg# = 0.42), just like chromian spinels of group IV. Chromian spinels of group VI also show the highest average concentrations of FeO (30.3%) and ZnO (0.13%) compared to the others.

### 4.7. Kiryushkino Paleoplacer

The Kiryushkino paleoplacer was discovered in a sandy gravel quarry 23.2 km south of the Sabantuy paleoplacer (Figure 1b). The area of the chromite placer is $\geq$200 m$^2$. The studied section (Figure 3) is represented by the following rock types: cross-bedded brownish gravel pebbles sharply substituted by fine- to medium-grained sandstones (0.9 m), gray cross-bedded medium- to coarse-grained sandstones (0.4 m), cross-bedded brownish gravel pebbles (0.7 m), grayish brown coarse-grained cross-bedded sandstones (0.3 m), brownish gray tabular limestones (0.4 m) and bluish gray siltstones with a 12 cm-thick interlayer of dark red mudstones (0.6 m). Horizontally bedded below are gray fine-grained sandstones

with undefined thickness, since only the surface of these sequences is exposed. The chromite-rich horizon was discovered in cross-bedded sandstones of the upper sequence. Here, we can define the upper 30 cm-thick and lower 10 cm-thick subhorizons of chromite sandstones represented by numerous fine interlayers (1–3 mm) (Figure 2d). The lower subhorizon is bedded 0.5 m below the upper one. Sandstones with chromite-rich interlayers are weakly cemented and their detailed petrography has not been provided. Sandstones 3 m below the chromite-bearing horizon contain well-sorted and medium-rounded grains. They have pore-filling carbonate cement. The average size of particles is 0.3 mm. The portion of rock clasts is 57–61% and the portion of silicate minerals is high (24%–26%), which makes them similar to sandstones from the Sukhoy Izyak paleoplacer. On the QRF diagram compositions of sandstones are displaced to the field of feldspatic litharenite, like chromite-rich sandstones of the Sukhoy Izyak placer (Figure 4). In cross-bedded sandstones below the chromite-bearing horizon, numerous remains (ferruginous) and imprints of stems and leaves of fossil plants were found.

Chromian spinels were selected from three combined samples of chromite sandstones (B$_{21}$-205, B$_{21}$-213 and B$_{21}$-215) from the upper and lower subhorizons. Chromian spinels account for 76% of the heavy mineral fraction. Out of 98 separated grains, 3 grains (3.1%) belong to group I, 16 grains (16.3%) to group II, 24 grains (24.5%) to group III, 38 grains (38.8%) to group IV, 12 grains (12.2%) to group V, 5 (5.1%) to group VI. They comprise both non- and rare well-rounded, as well as non- and highly corroded grains. In general, many grains in contrast to, for instance, the Verkhny-Yaushevo or Bazilevo paleoplacers show distinct crystallographic contours and a clear faceting. The size range of grains is 130–320 μm, the size of most grains (>70%) range from 180 to 200 μm. Crystals of group I are completely non-rounded (Figure 13a) or poorly rounded. Poorly rounded octahedrons with vicinals or flattened individuals are most common among grains of group II. Grains of group III encompass myrihedron, and the crystal transient from an octahedron to a dodecahedron or octahedron with facet distortions with the formation of trapezohedron-shaped grains (Figure 13b), as well as myriohedral crystals. Group IV comprises highly distorted elongated and flattened octahedron and dodecahedron, resembling rhombohedron in shape, and trigonal trapezohedron (Figure 13c,d), truncated tritetrahedra and tetragonal pyramids. Some of them may be weakly corroded fragments with even flat fracture presenting false facets. Conchoidal and isometric fragments are widespread among grains of group V. All three grains of group VI are isometric-shaped. In the Kiryushkino paleoplacer, like in the Novomikhaylovka paleoplacer, in contrast to the Kolkhoznyi Prud, Verkhne-Yaushevo, Sukhoy Izyak nad Bazilevo paleoplacers, the percentage of chromian spinel grains with a dense homogenous structure or at least with the solid core is significantly higher (~50%). Grains with finely mottled distribution of magnetite substituting chromian spinel are widespread among grains with an irregular inner structure.

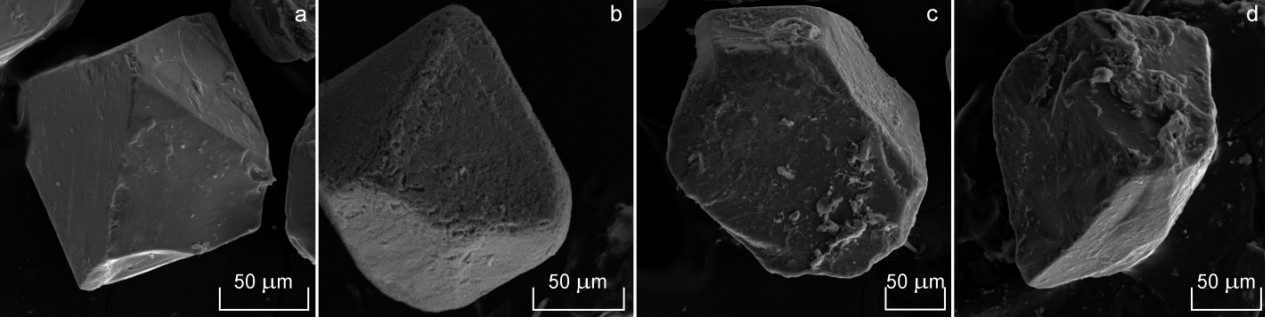

**Figure 13.** Morphology of chromian spinel grains from the Kiryushkino paleoplacer (SE images): (**a**)—non-rounded octahedron (group I), (**b**)—octahedron with distorted facets with the shape of a tetragonal trapezohedron (group III), (**c**)—distorted flat rhombohedron-shaped grain with pyramidal facets (group IV), (**d**)—distorted trigonal trapezohedron-shaped grain (group IV).

Chromian spinels from various morphogroups are closely approximated in terms of their average chemical composition (Table 2). This makes the Kiryushkino paleoplacer similar to the Sabantuy and Novomikhaylovka paleoplacers. The bulk of analyses complies with the chromite composition, a great number of points on the $Al^{3+}$–$Cr^{3+}$–$Fe^{3+}$ diagram fall into fields of alumochromite and subferrichromite (Figure 7h). Only one analysis of Ti-low chromite was obtained for group I. In contrast to the other placers, chromian spinels of group VI are the highest in Cr here ($Cr_2O_3$ 54.3 wt.% on average, Cr# = 0.83), while the ZnO content in them is below the detection limit, which differs from chromian spinels of group VI in the other studied placers. However, all grains are corroded. Chromian spinels of group V are the highest in Mg (MgO 12.0 wt.%, Mg# = 0.56), while chromian spinels of group III are the lowest in Mg (MgO 7.9 wt.%, Mg# = 0.39). Chromian spinels of group II are compositionally identical to group II of the Novomikhaylovka paleoplacer (Cr# = 0.74, Mg# = 0.43).

## 5. Discussion

### 5.1. Environment of Sedimentation

Metamorphic, sedimentary and magmatic complexes of the collisional orogeny contributed to the formation of the Middle Permian Molasse formation terrigenous deposits in the Pre-Urals [73]. The presence of ultramafic rocks in the provenance is evidenced not only by a high chromian spinel content of the sandstone heavy mineral fraction, but also by geochemical features of sandy and argillaceous rocks rich in Cr and Ni [73]. Our petrographic studies of the Kazanian sandstones showed that the bulk of clasts could be transferred from decomposed Riphean sedimentary-metamorphic complexes of the Bashkir Meganticlinorium [21]. The Early Permian Ural Foredeep had been filled in by the Early Biarmian (Guadalupian) epoch detrital material that was distributed further westwards simultaneously with the sea regression [69]. The terrigenous material was transported by large rivers oriented sublatitudinally [21,81]. The find of chromite paleoplacers in the Kazanian sediments clearly indicates an episodic exhumation in the provenance of chromite-bearing complexes potentially rich in chromite.

The structural and textural features of the studied sandstones indicate their sedimentation in the aquatic settings. The presence of sandy gravel sediments on the one hand and limestones on the other (Figure 3) in thin sections, as well as finds of brachiopods and vegetal detrital material (even large tree trunks in the Bazilevo section) suggest recurring fast interchanges of continental settings and shallow-marine and vice versa. The lithology of chromite-bearing sections is similar, but there is no identical section, which witnesses local differences of the sedimentation environment. Most of the studied placers (excluding the Sabantuy) show either unidirectional cross-bedding of chromite-rich sediments or subhorizontal bedding in thin rapidly pinching out lenses of sandstones. Sandy lenses are commonly wavy in the longitudinal section. Their grains are well-sorted and poorly rounded. These features are typical of alluvial deposits [82–84]. Cross- and gently undulated, locally multidirectional slanting bedding was indicated in sandstones of the Sabantuy section. The clasts are very well-sorted and well-rounded. Such features are typical of littoral (possibly, delta) sediments [82–84]. In addition, the Sabantuy paleoplacer differs from the others in the size of the chromite-rich bed. Alluvial placers are commonly smaller, while littoral placers are thicker and longer [22,85]. Chromite sandstones of the Sabantuy paleoplacer occur in rather homogenous polymictic sandstones with consistent bedding parameters in littoral placers, while chromite sandstones from other placers are bedded in rapidly pinching out wavy sequences. Coastal-marine placers of the Southern Carolina (Kiawah Island) [86], Western Maharashtra [25,26], etc. can be considered the present-day analogues of the Sabantuy paleoplacer in the shape of the ore bed and structural-textural features of country rocks, but not in mineralogy.

Chromite paleoplacers of the Southern Pre-Urals are unique, being not proximal, but distal placers, though they are not complex. Distal placers are produced by rewashing of great masses of ore-bearing sediments. Thus, they contain various ore minerals character-

ized by supergene stability (ilmenite, zircon, rutile, magnetite, chromite, etc.) [22]. Chromite is the main mineral in the Southern Pre-Ural paleoplacers, while ilmenite and especially zircon are minor. Most of the explored world placers, not only chromite-bearing, are young, while ancient mineral placers are poorly studied [87]. Chromite paleoplacers of the Southern Pre-Urals show features of a proximal placer [88], but neither the primary source is found nearby (at least within a radius of 200 km) nor any sign of its former presence. This is probably the first find of such distal placers in the world. One of the possible scenarios for explaining Pre-Urals placers is the involvement of tectonic processes, namely, the presence of an ophiolitic allochthon near the chromite placer formation zone [74]. However, this is only an assumption, not yet supported by factual material. We presume that the potential for discovering new chromite paleoplacers in the study area is not constrained and all of them can be referred to the new Southern Pre-Ural chromite-bearing ore region.

*5.2. Relationship between Morphology and Chemical Composition of Chromian Spinels*

Petrographic studies of sandstones from chromite-bearing sections showed that they had one common feeding province. Metamorphic and sedimentary rocks prevail in the composition of their clasts. Two sections, i.e., the Sukhoy Izyak and Kiryushkino, slightly differ in a greater amount of magmatic rock fragments in sandstones. Chromite-bearing rocks sometimes have been found in the source region. The morphological and geochemical variety of chromian spinels can indicate their heterogenous source. We defined six morphogroups of chromian spinels that occur in almost every placer. Notably, quantitative ratios of the morphogroups differ in various placers (Figure 14). The Sabantuy paleoplacer differs from the others in a greater number of octahedral grains (~60%) (regular, truncated, distorted). Fragmental and xenomorphic grains are minor in this placer. In the current study, such grains were not even included in the selection, although they were previously found [21,74]. Morphogroups of the Novomikhaylovka and Kiryuskino paleoplacers are the closest to the Sabantuy paleoplacer. They show rather a great portion of octahedral grains (20%–30%) compared to the rest placers, where the share of octahedral crystals is <10%. The Kolkhoznyi Prud, Verkhne-Yaushevo, Sukhoy Izyak and Bazilevo paleoplacers are similar in their high content of non-octahedral, fragmental and xenomorphic grains. The bulk chemical composition of chromian spinels in all placers is close (Figure 7a), which indicates same sources for chromian spinels in all placers. Judging by the varied composition of the morphogroups, only the contribution of certain sources for chromian spinels differs. We can assume that the formation of placers with a high portion of octahedral grains some differed from that of placers highly deficient in octahedral grains by the age. Thus, comparison of chromite-bearing rocks according to dominant forms of chromian spinel grains can be a useful tool for stratigraphic correlation and division of the sediments.

As shown above, the studied detrital chromian spinels vary in Mg# and Cr#, but most analyses (>70%) belong to the moderate range of Mg# = 0.3–0.6 and Cr# = 0.6–0.8. On the Mg#–Cr# diagram, most of the points correspond to the trend of negative correlation between Mg# and Cr# and fall on the field of fore-arc peridotites (Figure 15a). The $TiO_2$–Cr# diagram shows that there is no significant correlation between $TiO_2$ and Cr#, and many points are displaced from the fields of fore-arc peridotites and abyssal peridotites (Figure 15c). In the $Al_2O_3$–$TiO_2$ diagram, the bulk of analyses (~70%) falls within the field of boninitic chromian spinels. However, all these features do not contradict with the ophiolitic nature of the detrital chromian spinels, since in all indicated diagrams, their compositions completely or significantly overlap with the compositions of chromian spinels from peridotites, dunites and chromitites of the Kraka ophiolitic massif.

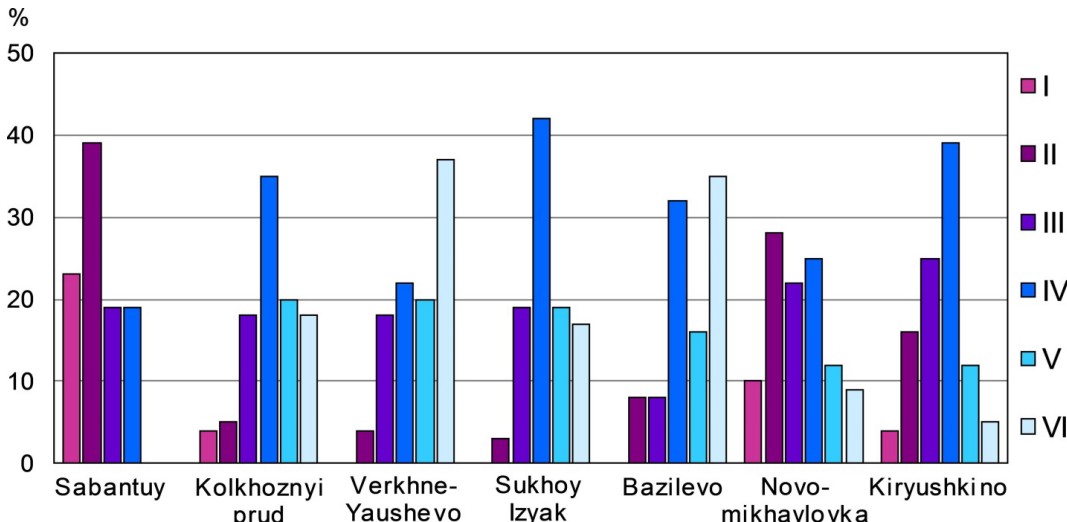

**Figure 14.** Histograms for morphological groups of chromian spinels from chromite paleoplacers in the Southern Pre-Urals.

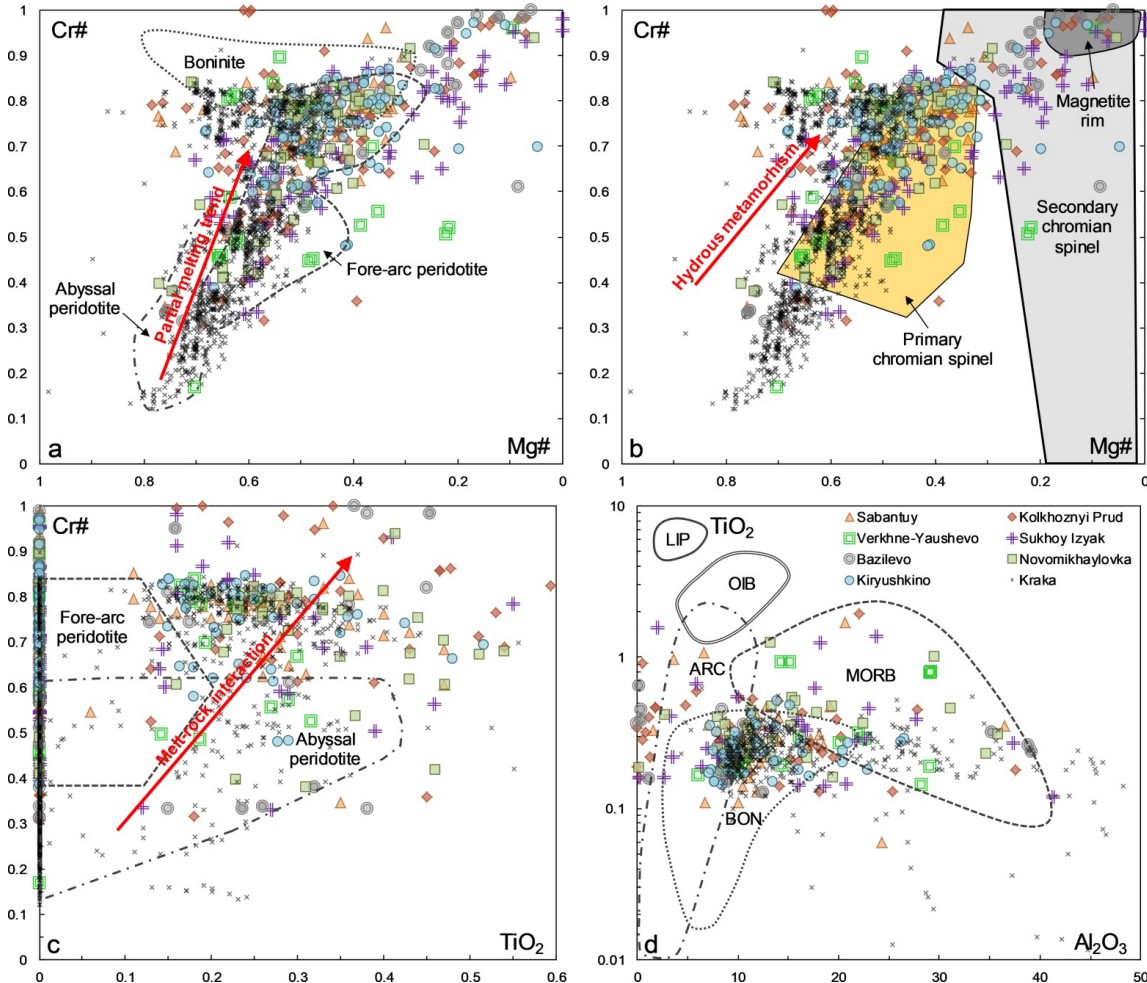

**Figure 15.** Discrimination diagrams for chromian spinels from the studied paleoplacers and Kraka ophiolitic massif (peridotites, dunites and chromitites): (**a**)—Mg#–Cr# (mol.%) with fields, after [89,90], (**b**)—Mg#–Cr# (mol.%) with fields, after [91], (**c**)—TiO$_2$–Cr# (wt.%–mol.%) with fields, after [89], (**d**)—Al$_2$O$_3$–TiO$_2$ with fields, after [92]. ARC = arc volcanites, BON = boninites, LIP = large igneous province, MORB = mid-ocean ridge basalts, OIB = ocean island basalts.

Chromian spinels are resistant to the mechanical impact and commonly preserve their crystallographic contours even in the tidal zone [93]. The detailed morphological study of mineral indicators of kimberlites, where chromian spinel plays an important role, revealed that the degree of mechanical wear of grains in alluvial environments is negligible [94]. At the same time, in the littoral environment, the mechanical wear of grains is already clearly expressed [95]. In most of the studied placers, chromites are almost not rounded, and only the Sabantuy paleoplacer contained poorly rounded grains. It provides further evidence to the littoral genesis of the Sabantuy paleoplacer and to the alluvial genesis of the other paleoplacers.

Chromian spinel is an accessory mineral in many types of igneous rocks, but forms and sizes of crystals vary [43,45,54,55,57]. Volcanites commonly contain small octahedral crystals of chromian spinels with the size of up to 50 μm, while they are more than twice bigger in intrusive rocks with the similar bulk composition [44,96]. The size ranging of chromian spinel grains in the studied placers ranges from 150 to 250 μm. Therefore, the presence of accessory chromian spinels from intrusive rocks (e.g., peridotites, gabbro) in the studied placers is more possible than the presence of those from volcanites (picrites, basalts). On the $Al^{3+}$–$Cr^{3+}$–$Fe^{3+}$ and $Al_2O_3$–$TiO_2$ diagrams, compositional fields of the studied chromian spinels overlap with the field of MORB only partly (Figures 7i and 15d).

Morphogroups II and III of the studied placers include single grains that morphologically and chemically comply with chromian spinels from diamond-bearing kimberlites. Myriohedral and octahedral grains with vicinals, showing >62 wt.% $Cr_2O_3$, <7.5 wt.% $Al_2O_3$ and <0.5 wt.% $TiO_2$, can be referred to them [20,40,42]. Hence, we cannot exclude that chromian spinels from diamond-bearing kimberlites could be present in the studied placers. Indeed, finds of detrital diamonds are noted in the Urals, including in the Southern Urals [71]. Although kimberlites have not been found. At the same time, we note that most myriohedral grains of chromian spinel are characterized by moderate chromium content ($Cr_2O_3$ 50–60 wt.%).

The low number of well-rounded (spherical) chromian spinel grains can testify to their origin from more ancient chromite-bearing sedimentary sequences.

In general, the prevalence of chromian spinel over other ore minerals (magnetite, titanomagnetite, ilmenite, etc.) in placers matches the ophiolitic nature of detrital chromian spinels best of all [74]. Most studied grains (>70%) on the $Al^{3+}$–$Cr^{3+}$–$Fe^{3+}$ diagram occupy fields of chromite and alumochromite, mainly reflecting variations of Cr and Al, and they are almost completely overlapped by compositional fields of chromian spinels from the Kraka ophiolitic massifs (Figure 7i). The octahedral habitus of detrital chromian spinels does not contradict with their ophiolitic origin. Myriohedral shape of chromian spinel grains also does not contradict the ophiolitic nature of the source. In general, chromian spinels from ophiolitic complexes are probably the most diverse in morphology compared to other formations. Combined with geochemical criteria, this fact suggests there is no need to attract any other sources for detrital chromian spinels, but ophiolitic. The ternary diagram also shows a compositional field of chromian spinel from ophiolitic chromitites, which are the chromite-richest ores (Figure 7i). The presence of chromitites in the provenance obviates the need for great masses of washed material to produce chromite placers. Processing of great masses of chromite-bearing rocks must have left evidence as fragments of dunites and lerzolites in sandstones or as specific weathering crusts in the Permian sections. However, only minor serpentinite clasts occur in the studied chromite-bearing sandstones (<1%). The absence of weakly stable minerals, such as olivine and pyroxene, and fragments of unstable rocks, such as dunites, lerzolites and harzburgites, provides further proof to the affinity of our placers to distal type.

Fragmental, elongated and flattened forms with parallel rhombohedron-shaped facets are quite common. Such grains can be disintegration products of ophiolitic chromitites that show various forms of chromian spinel grains, in particular, intensely fractured aggregates (Figure 16). Presumably, these aggregates were destroyed during their transfer and the grains were sorted in size. Thus, we may observe pseudorhombohedral, trape-

zohedral and other crystal forms of lower crystallographic systems. These grain forms may be not original, but only fragments of larger crystals with flat surfaces and regular crystallographic contours.

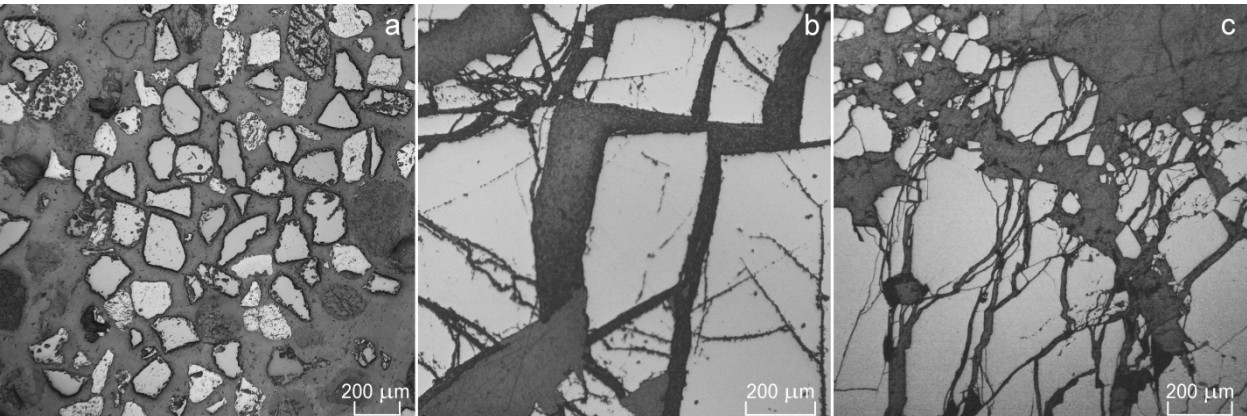

**Figure 16.** Optical images (reflected light) of Sabantuy chromite sandstone (**a**) and Kraka chromitites: (**b**)—Shatran mine, (**c**)—Bolshoy Bashart mine.

Sandstones with prevalent carbonate composition of their cement show a minor portion of corroded chromian spinel grains. Highly corroded grains of chromian spinels are distributed in sandstones having clayey and ferriceous-clayey cement that substituted carbonate cement. In these sandstones, chromian spinels have a loose porous structure and they are often substituted by chrommagnetite and occasionally by magnetite and hematite. The pores are filled with low-temperature silicates. Corroded grains of chromian spinels commonly show increased values of the ZnO content (up to 1.3 wt.%), which is typical of supergene alteration of chromian spinels [66,97]. Our observations indicate that xenomorphic grains of chromian spinels are the least resistant to corrosion; grains with a loose surface can be found among them (Figure 11d).

On the Mg#–Cr# diagram, most of the points fall into the field of primary chromian spinel (Figure 15b). However, fairly many points are displaced in the field of secondary chromian spinel, which may indicate a supergene alteration of some studied grains. Thus, highly corroded grains of chromian spinels partly or completely lose their indicative value for provenance.

### 5.3. Inclusions and Their Interpretation

Primary inclusions of three types, i.e., monomineral (olivine, clinopyroxene, orthopyroxene and amphibole), polymineral and glass, were found in chromian spinels of the Sabantuy paleoplacer. The composition of minerals from monomineral inclusions completely complies with ophiolitic peridotite [21]. The correlation of Mg# in olivine from inclusions and Cr# in host chromian spinel indicated that all points were distributed along the left side of the OSMA field. We explained the shift of olivine to a more Mg#-high composition area by the re-equilibration reaction between chromian spinel and olivine noted in peridotites and chromitites [98].

Glass inclusions occur in the un-, partly and completely recrystallized form. Polymineral inclusions represented by intergrowths of clinopyroxene, plagioclase, amphibole and secondary silicates can be completely crystallized melt inclusions, though subject to secondary alterations [21]. Figure 17 presents classification diagrams for bulk compositions of partly and completely crystallized inclusions and compositions of the volcanic glass. On the TAS-diagram their compositional points fall in fields of basalt and basaltic andesite (Figure 17a), while on the AFM diagram, they are plotted closer to the tholeiitic trend (Figure 17b). It underlines the proximity of melt inclusions and glasses to MORB [99]. Chromian spinels with these melt inclusions are high in Mg (Mg# = 0.7–0.8) and low in Cr (Cr# = 0.3–0.4), which totally complies with the composition of MORB chromian spinels [59,61]. Melt inclusions

with similar compositions and forms were found and studied in basic dikes of the Stravage ophiolitic complex (Albania) cutting pillow lavas of MOR [100].

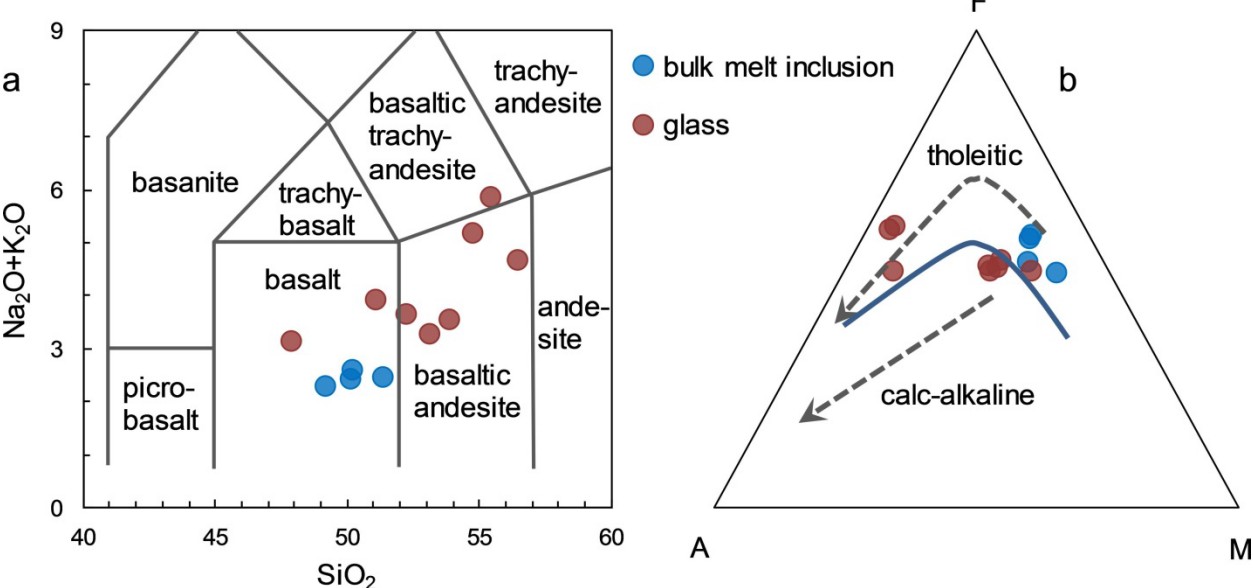

**Figure 17.** Classification diagrams for melt inclusions captured by chromian spinels from the Sabanuy paleoplacer: (**a**)—TAS (wt.%), after [101], (**b**)—AFM (wt.%), after [102].

As shown above, fragments of volcanites were found in sandstones comprising chromite-rich layers. Most of these fragments were altered by metasomatic and supergene processes, but we managed to define the composition of plagioclase phenocryst, i.e., $An_{63-34}$ (range from labrador to andesine), in the best-preserved samples. It proves the idea that volcanite clasts are product of destruction of basalts and andesibasalts that could be sources for detrital chromian spinels, particularly regular octahedral grains, in the studied placers. Both basalt fragments that are larger in size (0.3 mm) and chromian spinel octahedral crystals with smaller grains (0.2 mm) can have a single source. The density of chromian spinel is approximately 1.5 times higher than that of basalts, while their masses are nearly the same for the above sizes. Therefore, they have close hydrodynamic features in the sedimentation zone.

The found melt inclusions indicate that chromian spinels from the MORB-association are present in the studied placers. Typical features of these chromian spinels are an octahedral crystal shape and a high $TiO_2$ content (~0.5 wt.%). Basalts and gabbroids of MOR compose the upper section of the ophiolitic association and occur as product of crystallization of primitive magmas [72,103]. Thus, even Ti-high octahedral chromian spinels with melt inclusions do not contradict the ophiolitic nature of chromian spinels from paleoplacers in the Southern Pre-Urals. There is no need to attract additional sources for chromian spinels, except ophiolitic, to explain these placers' formation.

## 6. Conclusions

In addition to the earlier discovered Sabantuy paleoplacer, six minor chromite paleoplacers were discovered in the Kazanian-stage sandstones in the Southern Pre-Urals. In most placers, chromian spinel is a head mineral of the heavy mineral fraction, while a number of lithological and petrographic features indicates affinity of these placers to distal type. This emphasizes the unique nature of these deposits, since the world knows no analogues of such placers. The potential of discovering new chromite paleoplacers in the studied region is not constrained, and so far, they can be united into the new Southern Pre-Ural Cr-bearing region.

　　　　Petrographic studies of Cr-bearing sandstones indicate a similar source for detrital material in all sections. The structural and textural features of the rocks showed that all new finer paleoplacers have the alluvial genesis, while the Sabantuy paleoplacer was formed in the littoral setting.

　　　　Chromian spinels in the studied placers are morphologically diverse, which, however, is not evidence of the heterogeneity of their source. It is shown that regular octahedral, distorted octahedral, myriohedral and combinational polyhedral, as well as xenomorphic grains, can be of ophiolitic origin. The placers differ in prevalent forms of chromian spinels. The Sabantuy, Novomikhaylovka and Kiryushkino paleoplacers are similar in their high content of octahedral chromian spinel grains, while the Kolkhoznyi Prud, Verkhne-Yaushevo, Sukhoy Izyak and Bazilevo paleoplacers mainly contain non-octahedral, fragmental and xenomorphic chromian spinel grains. Studying the prevalent form of chromian spinel grains can be useful for stratigraphic correlation and division of sandy sediments.

　　　　As for the bulk chemical composition, chromian spinels from all paleoplacers are similar and almost completely overlap with chromian spinels from the Kraka ophiolitic massif in the Southern Urals. The pooled morphological and chemical data on chromian spinels, as well as on compositions of inclusions in them, indicate that the ophiolitic source is fairly enough to explain the placers formation.

**Supplementary Materials:** The following supporting information can be downloaded at: https://www.mdpi.com/article/10.3390/min12070849/s1, Table S1: EM.

**Author Contributions:** Sampling, investigations, analytical results and interpretation, I.R.R. and D.E.S.; analytical investigations, M.A.R.; petrographic study, A.A.S.; Writing paper, I.R.R. All authors have read and agreed to the published version of the manuscript.

**Funding:** This research was funded by the Council of the President of the Russian Federation, grant number MK-857.2021.1.5 and grant RB NOC-GMU-2021 (I.R.R.). The analytical studies were supported by RSF, grant number 22-17-00019 (D.E.S.) and State Contract of IG UFRC RAS no. FMRS-2022-0012 (M.A.R. and A.A.S.).

**Data Availability Statement:** The data presented in this study are available on request from the corresponding author.

**Acknowledgments:** We are grateful to T.A. Miroshnichenko for the English translation of our manuscript.

**Conflicts of Interest:** The authors declare no conflict of interest.

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
