# Peer review of "Chromian Spinels from Kazanian-Stage Placers in the Southern Pre-Urals, Bashkiria, Russia: Morphological and Chemical Features and Evidence for Provenance"

_minerals, doi:10.3390/min12070849_

Round 1

Reviewer 1 Report

MS Description & Synopsis

On the basis of the morphological features of detrital chromian spinels and their chemical composition, the authors provide constraints on sources for seven chromite placers found in sandy sediments of the Kazanian stage of the Permian System in the Southern Pre-Urals.

Presentation & Scientific Interpretation

The text is generally well-written. As well, the main text is very long and I think can be considerably shortened or portions moved to the Supplementary Document, for example the full description of seven placer listed in part 4.

Other than the above, I have few issues with the MS.

(1) please state clearly the scientific problem that will be focused on in this study and can be resolved by your data in the INTRODUCTION part. The language should be concise.

(2) please show bullet points that capture the novel results of your research in the CONCLUSION part. For example, line 1005-1015 is a simple repetition of the previous part (line 191-199).

(3) authors suggest that the source for chromite placers is the Kraka ophiolite complex in the Southern Urals. However, the Kraka massif locates more than 150km east of the deposits according to Fig. 1. Why the placers appear in present position not other closer place between the two? Whether chromitites exist in the Kraka ophiolite is not discussed, nor is the tectonic setting of the ophiolite.

(4) Why compare the studied chromites with chromian spinel from diamond-bearing kimberlites? Whether diamond-bearing kimberlites are present in research area and, if so, they need to be described appropriately.

Overall, the manuscript is good and deserves publication in Minerals after major revision.

Author Response

Reviewer 1

 Thank you very much for the review, which allowed us to improve the text of our article. All your comments have been taken into account. The text has been carefully edited, all changes can be tracked in the minerals-1765784 — marks.docx version.

Below are some answers to questions and comments.

The text is generally well-written. As well, the main text is very long and I think can be considerably shortened or portions moved to the Supplementary Document, for example the full description of seven placer listed in part 4.

Reply We appreciate your comment. However, we believe that it will be more convenient for readers if the description of all studied objects is present in the text, and not in supplementary materials. This description is important for substantiating the genesis of sandstones. We have tried to shorten the description in places and make it more concise.

 (1) please state clearly the scientific problem that will be focused on in this study and can be resolved by your data in the INTRODUCTION part. The language should be concise.

Reply We added a more precise description of the scientific problem in the Introduction Chapter.

(2) please show bullet points that capture the novel results of your research in the CONCLUSION part. For example, line 1005-1015 is a simple repetition of the previous part (line 191-199).

Reply Done. We improved this part.

(3) authors suggest that the source for chromite placers is the Kraka ophiolite complex in the Southern Urals. However, the Kraka massif locates more than 150km east of the deposits according to Fig. 1. Why the placers appear in present position not other closer place between the two? Whether chromitites exist in the Kraka ophiolite is not discussed, nor is the tectonic setting of the ophiolite.

Reply We discussed this in the text (chapter 5.1). Not necessarily the Kraka massif itself, but a massif similar to it could be the source. We have corrected this point. The tectonic nature of the Kraka ophiolitic massif is also mentioned in chapter 2. Chromitite bodies are present in Kraka massif and are mentioned in the text. Moreover, we made a comparison with chromitites in chapter 5.2.

(4) Why compare the studied chromites with chromian spinel from diamond-bearing kimberlites? Whether diamond-bearing kimberlites are present in research area and, if so, they need to be described appropriately.

Reply Findings of detrital diamonds are indeed noted in the Urals. We cannot ignore the potential presence of diamondiferous kimberlites in the source, since the presence of myriohedral chromian spinel crystals is one of the hallmarks of diamondiferous kimberlites [Kaminsky et al., 2004; Kaminsky, Geo, 2006; Grakhanov et al., 2022]. However, we have improved this point.

Reviewer 2 Report

The paper uses wrong nomenclature for many of the technical terms. Although I have not completed editing the paper in detail, the authors can use the existing comments in the pdf as a guide for the remainder of the paper.

The biggest flaw is the mineralogy. Do not describe crystals as orthorhombic or tetragonal - They are isometric with some faces overdeveloped. use hkl parameters to describe crystal forms.

Author Response

Reviewer 2

 Thank you very much for the detailed review, which allowed us to improve the text of our article. All your comments have been taken into account. The text has been carefully edited, all changes can be tracked in the minerals-1765784 — marks.docx version.

Below are some answers to questions and comments.

I am unable to find the meaning of this term. Mistranslation? Massive/multigranular?

Reply The term "myriohedral crystal" is widely used to describe ore minerals (chromite, picroilmenite) of diamond-bearing kimberlites. There are crystals with many extra facets that form the appearance of a rounded grain. Myriohedral is not rounded. We believe this term is successful. Some references to works in which the term "myriohedral crystal" is used.

https://doi.org/10.2113/RGG20214431

http://www.geologyontario.mndm.gov.on.ca/mndmfiles/afri/data/imaging/20000001165/20002145.pdf

https://doi.org/10.1016/j.lithos.2004.03.035

What is this? Stick with % Cr2O3 / What is this number?

Reply Chapter 3: Indicative geochemical values were estimated using cations 218 Cr#=Cr/(Cr+Al), Mg#=Mg/(Fe2++Mg), Fet=Fe2++Fe3+.

The Cr# value is widely used in the chromian spinel geochemistry. Also Mg#.

Round 2

Reviewer 1 Report

Overall, the revised manuscript is good and deserves publication in Minerals after minor revision. Here are few issues. 

(1)   Line 46-47. “Chromite placers are divided into three genetic types, i.e. colluvial, alluvial and littoral [22].”However, the authors use the terms of the previous version. For example, Line 13, Line 55, and Line 60. The authors should carefully check the full text and standardize the terminology.

(2)   Please add references at line 63, line 66.

(3)   Line 108. “spinels richest”or “Cr-rich”?

Author Response

Thanks for your helpful comments.

Overall, the revised manuscript is good and deserves publication in Minerals after minor revision. Here are few issues.

(1)   Line 46-47. “Chromite placers are divided into three genetic types, i.e. colluvial, alluvial and littoral [22].”However, the authors use the terms of the previous version. For example, Line 13, Line 55, and Line 60. The authors should carefully check the full text and standardize the terminology.

Reply Ok. We checked all the text and corrected the terms (marked by the green).

(2)   Please add references at line 63, line 66.

Reply Here are refers to the literature source [31 and 32 according to the list].

(3)   Line 108. “spinels richest”or “Cr-rich”?

Reply Spinels richest in Cr, it's relative.

This manuscript is a resubmission of an earlier submission. The following is a list of the peer review reports and author responses from that submission.

Round 1

Reviewer 1 Report

The research is sounding and deserves publication in Minerals.

Author Response

Dear reviewer,

Thank You for positive report.

Reviewer 2 Report

The manuscript is impressive. I don't think any major corrections are required apart from some English punctuations if authors wish. 

Author Response

Dear reviewer,

Thank You for positive report. We’ve made the style and grammar check by a skilled English expert.

Reviewer 3 Report

Lines 93-94: “Ophiolites most typically show chromite, alumochromite, chrompicotite and picotite, while chromites from ophiolites commonly contain more chromous chromspinels, i.e. alumochromite and chromite low...” is ambiguous. Rephrase it.

Introduction: the authors may want to be more precise to tell what the goal of this paper is and how they think can get it.

Very nice and complete review of the chromite types and their genetic affiliation in the second subchapter.

I would like the authors to explain how they discriminated between Fe2+ and Fe3+ when they determined the quantitative composition of the chromites (Indicative geochemical values were estimated using cations Cr#=Cr/(Cr+Al), Mg#=Mg/(Fe2++Mg), Fet=Fe2++Fe3+). Was total Fe as Fe3+ (they used it in the diagrams from page 14. If yes, how did you calculated the Fe#? I looked at the supplemental data and I see only the Mg#, Cr#, and Fe#, but no the atoms per formula unite (apfu). You said “…an automatic mode with factory standards applied (synthetic and natural compounds).” What are the standards?

Can you explain/define what “removal area” is (line 742)?

Author Response

Dear reviewer,

Thank You for positive report. Responses to comments are contained in a special file.

Reviewer 4 Report

The manuscript entitled "Chromspinels from Kazanian-stage places in the Southern Pre-Urals, Bashkiria, Russia: typomorphical and typochemical features and constraints to source unraveling" presented morphological and chemical compositions of chromite sampled from six chromite placers. This work is regional and I do not think it has any significant geological implications. Main comments include: (1) There are many figures about morphological classification of chromite. Please explain why six types of morphological features are defined. The authors try to find the relationship between morphology features of chromite and their chemical compositions, but they overlap a lot without any clear trend. (2) Chemical compositions of chromite were determined in order to identify their sources. Figure 7 shows that a lot of data were in the MORB field, but the authors still get the conclusion that chromites have ophiolite nature, which seems not to be supported by the evidence provided in the manuscript.

Author Response

Dear reviewer,

Thank You for report. Responses to comments are contained in a special file.

Round 2

Reviewer 4 Report

The manuscript entitled "Chromspinels from Kazanian-stage places in the Southern Pre-Urals, Bashkiria, Russia: typomorphical and typochemical features and constraints to source unraveling" presented morphological and chemical compositions of chromite sampled from six chromite placers. This work is regional and I do not think it has any significant geological implications. The authors claimed that their purpose is to find the relationship between the morphology and chemical compositions of chromites, and the results show that they have no association. Obviously, the morphology of chromite is mainly influenced by mechanical impacts, which did not influence their chemical compositions due to low temperatures. Therefore, I do not think this work has any significant implications.